# 3D integration enables ultralow-noise isolator-free lasers in silicon photonics

Chao Xiang[1,2,7 ✉], Warren Jin[1,3,7], Osama Terra[1,6,7], Bozhang Dong[1,7], Heming Wang[1], Lue Wu[4], Joel Guo[1], Theodore J. Morin[1], Eamonn Hughes[5], Jonathan Peters[1], Qing-Xin Ji[4], Avi Feshali[3], Mario Paniccia[3], Kerry J. Vahala[4] & John E. Bowers[1,5 ✉]

Photonic integrated circuits are widely used in applications such as telecommunications and data-centre interconnects[1–5]. However, in optical systems such as microwave synthesizers[6], optical gyroscopes[7] and atomic clocks[8], photonic integrated circuits are still considered inferior solutions despite their advantages in size, weight, power consumption and cost. Such high-precision and highly coherent applications favour ultralow-noise laser sources to be integrated with other photonic components in a compact and robustly aligned format—that is, on a single chip—for photonic integrated circuits to replace bulk optics and fibres. There are two major issues preventing the realization of such envisioned photonic integrated circuits: the high phase noise of semiconductor lasers and the difficulty of integrating optical isolators directly on-chip. Here we challenge this convention by leveraging three-dimensional integration that results in ultralow-noise lasers with isolator-free operation for silicon photonics. Through multiple monolithic and heterogeneous processing sequences, direct on-chip integration of III–V gain medium and ultralow-loss silicon nitride waveguides with optical loss around 0.5 decibels per metre are demonstrated. Consequently, the demonstrated photonic integrated circuit enters a regime that gives rise to ultralow-noise lasers and microwave synthesizers without the need for optical isolators, owing to the ultrahigh-quality-factor cavity. Such photonic integrated circuits also offer superior scalability for complex functionalities and volume production, as well as improved stability and reliability over time. The three-dimensional integration on ultralow-loss photonic integrated circuits thus marks a critical step towards complex systems and networks on silicon.

Following the path of electronic integrated circuits (EICs), silicon (Si) photonics holds promises to enable photonic integrated circuits (PICs) with high densities, advanced functionality and portability. Although various Si photonics foundries are rapidly developing PIC capabilities—enabling volume production of modulators, photodetectors and most recently lasers—Si PICs have yet to achieve the stringent requirements on laser noise and overall system stability imposed by many applications such as microwave oscillators, atomic physics and precision metrology[9–11]. Semiconductor lasers must strongly suppress amplified-spontaneous-emission noise to achieve narrow linewidth for these applications[12]. They will also require isolation from the rest of the optical system, otherwise the laser source will be sensitive to back-reflections from downstream optical components that are beyond the control of the PIC designer[13]. In many integrated photonic solutions, a bulk optical isolator must be inserted between the laser chip and the rest of the system, significantly increasing the complexity, as well as the cost of assembly and packaging[14].

To enrich the capabilities of Si PICs and avoid multi-chip optical packaging, non-group-IV materials need to be heterogeneously integrated to enable crucial devices, including high-performance lasers, amplifiers and isolators[15–17]. It has now been widely acknowledged that group III–V materials are required to provide efficient optical gain for semiconductor lasers and amplifiers in Si photonics regardless of the integration architecture, but concerns still remain for a complementary metal–oxide–semiconductor (CMOS) fab to incorporate magnetic materials, which are currently used in industry-standard optical isolators[18].

Fortunately, a synergistic path towards ultralow laser noise and low feedback sensitivity exists—using ultrahigh-quality-factor ($Q$) cavities for lasers that not only reduce the phase noise but also enhance the feedback tolerance to downstream links. These effects scale with the cavity $Q$ and ultrahigh-$Q$ cavities would thus endow integrated lasers with unprecedented coherence and stability[19,20]. The significance is twofold. First, the direct integration of ultralow-noise lasers on Si PICs without

[1]Department of Electrical and Computer Engineering, University of California, Santa Barbara, Santa Barbara, CA, USA. [2]Department of Electrical and Electronic Engineering, The University of Hong Kong, Hong Kong, China. [3]Anello Photonics, Santa Clara, CA, USA. [4]T. J. Watson Laboratory of Applied Physics, California Institute of Technology, Pasadena, CA, USA. [5]Materials Department, University of California, Santa Barbara, Santa Barbara, CA, USA. [6]Present address: Primary Length and Laser Technology Lab, National Institute of Standards, Giza, Egypt. [7]These authors contributed equally: Chao Xiang, Warren Jin, Osama Terra, Bozhang Dong. ✉e-mail: cxiang@eee.hku.hk; bowers@ece.ucsb.edu

the need for optical isolators simplifies PIC fabrication and packaging. Furthermore, this approach does not introduce magnetic materials to a CMOS fab as isolators are not obligatory for such complete PICs.

## 3D integration of lasers and ultralow-loss PICs

Now we consider developing an integration architecture and process flow to seamlessly integrate the III–V-based lasers with ultralow-loss (ULL) waveguides. Among various ULL integrated photonics platforms, silicon nitride (SiN) has emerged as the leading performer and enabled a series of breakthroughs in metrology, sensing and telecommunications[21–23]. To achieve ultralow waveguide loss, SiN waveguides require high-temperature annealing[24–27] that violates the thermal budget of back-end-of-line semiconductor manufacturing processes. Front-end-of-line-fabricated ULL SiN waveguides are nonetheless susceptible to subsequent processing steps that could introduce additional loss, especially during heterogeneous laser integration, which involves multiple etches and depositions. To address these issues, we propose to use three-dimensional (3D) structures for the integration of lasers with ULL waveguides. Recent years have witnessed the development of 3D integration in electronics by heterogeneously or monolithically integrating layers for increased circuit densities and functionalities[28,29]. In photonics, 3D integration has been investigated for monolithic devices (for example, waveguides, modulators and photodetectors (PDs))[30] and heterogeneously integrated lasers[31]. Here we combine monolithic and heterogeneous 3D integration to fully unlock the potential of enabling complex and high-performance photonic devices and integrated circuits.

We effectively separate a 3D Si PIC into layers with respective photonic functionalities, as shown in Fig. 1a. The designed device consists of four major functionality layers, including a III–V gain layer, a Si PIC layer, a SiN redistribution layer (RDL) and a SiN ULL layer. The separation of the Si and the ULL SiN layers is approximately 4.8 μm, such that the ULL SiN layer can be effectively isolated from subsequent Si and indium phosphide (InP) processing steps, thereby retaining the performance of the ULL SiN (Extended Data Figs. 1–3). Such design necessitates interlayer transition across multiple functional layers. Unlike EICs that rely on the interlayer metallic vias for interconnects, 3D PICs leverage evanescent coupling across multiple layers and use waveguide geometry designs to achieve interlayer transitions that are otherwise forbidden. More specifically, we introduce a photonic RDL between the Si and ULL SiN layers for the control of coupling between the top active layers and the bottom ULL passive layer. A highly efficient, active–passive layer transition can be provided by the RDL where necessary (Extended Data Figs. 1 and 2).

The cross-section of the 3D PICs is also illustrated in Fig. 1a, showing its compatibility with foundry-available Si photonic components, including Si modulators and Si/germanium (Ge) PDs. In addition, such PICs could be further heterogeneously integrated with EICs for high-density 3D E-PICs. In our 3D photonic integration structure, the thick oxide separation forms an effective barrier for back-end loss origins such that ultrahigh-$Q$ resonators (with intrinsic $Q \approx 50$ million at the laser wavelength) are fully integrated with high-performance III–V/Si distributed-feedback (DFB) lasers (Fig. 1b). It must be noted that the 3D integration can result in multiple overlapping but decoupled photonic functionality layers—a goal not possible in previous demonstrations of heterogeneous integration[31,32]. This decoupling is now enabled by the large vertical mode separation, which is bridged by the SiN RDL. The multilayer structure of the fabricated device and InP epi wafer stack are shown in Fig. 1c.

## Single-chip self-injection locked lasers

We leverage self-injection locking of InP/Si DFB lasers to thermally tunable SiN ultrahigh-$Q$ resonators for ultralow-noise lasers on the 3D Si PIC. The working principle of such a device is summarized in Fig. 2a, which requires the laser and ring resonance wavelengths to match in the frequency domain, as well as the forwards and backwards signals to phase match in the time domain. To set the device to the proper working conditions, the InP/Si laser wavelength is tuned by the applied gain current, the SiN ring resonance is tuned by the thermal heater, and the forwards and backwards phases are tuned by the thermal phase tuner placed on the Si waveguides. Once both wavelength and phase-matching conditions are achieved, the free-running laser locks to the ultrahigh-$Q$ resonator owing to Rayleigh backscattering, resulting in several resonator-defined laser properties (Extended Data Figs. 4 and 5).

We investigate the dynamics and performance of the self-injection locked (SIL) laser using the measurement set-up shown in Fig. 2b. Owing to the availability of an on-chip phase tuner between the laser and ring resonator, we can clearly unveil the phase-dependent locking dynamics. In previous butt-coupled SIL experiments, tuning the chip-to-chip phase also varies the coupling loss, that is, the output power. Because the InP/Si laser and SiN resonator are heterogeneously integrated together, and the phase is thermally tuned on the chip, these are now decoupled in our experiment. Figure 2c shows the dependence of laser coherence on the phase-tuner power causing the phase shift. The laser wavelength is preset to match one of the ring resonances. We can observe a periodic dependence of the laser coherence when the laser-to-resonator phase is tuned by several one-direction π periods. Within each period, the laser goes through low-phase-noise locking, chaotic coherence collapse and high-phase-noise free-running regimes. These regimes are also observed from the time-domain power trace recorded on an oscilloscope when the current on the phase tuner is swept across a full period (Fig. 2c, bottom).

The ring resonance is another degree of freedom to control the locking dynamics. By tuning the thermal tuner current on the ULL SiN ring in both directions, the laser can be switched between the free-running state and the locked state as plotted in Fig. 2d (top). Depending on the phase, the locking range can be different for the bidirectional sweep. We observed about 1.4-GHz and 2.4-GHz locking ranges for the bidirectional sweep. This measured locking range is also affected by the thermal crosstalk during the resonance tuning, as evidenced by the laser frequency shift at free-running state. Figure 2d (bottom) shows the modelled asymmetric locking range without the thermal crosstalk at phase-matched conditions. Details of the calculation can be found in Supplementary Section V.

Ultralow laser frequency noise enabled by self-injection locking has been extensively studied in recent years[33]. These demonstrations, however, mostly rely on individual ultrahigh-$Q$ resonators, including crystalline whispering-gallery-mode resonators[34] and SiN ring[35] or spiral resonators[36]. The laser and the resonator are thus separate and need free space or fibre coupling. We recently demonstrated the self-injection locking of lasers with dispersion-engineered resonators on a heterogeneous chip for soliton microcomb generation[32]. However, the laser frequency noise is still relatively high, especially in the range of 1 kHz to 100 kHz, which is critical to microwave and sensing applications[37]. Our current device, with around 0.5 dB m⁻¹ ULL integrated with lasers on the same chip, showed the lowest laser frequency noise for a single-chip device, with around 250 Hz² Hz⁻¹ and 2.3 Hz² Hz⁻¹ at 10-kHz offset and at the white noise floor, respectively, for the through port. The white noise floor for the drop port is even lower (1.7 Hz² Hz⁻¹), showing about 5-Hz fundamental linewidth. It needs to be noted that these results are achieved with a relatively compact 30-GHz free spectral range (FSR) resonator, and the laser frequency noise is limited by the thermorefractive noise. Using a larger ring radius or a spiral-shaped resonator to reduce the thermorefractive noise, lower frequency noise (for example, subhertz fundamental laser linewidth) should be achieved using the same design strategy and fabrication process.

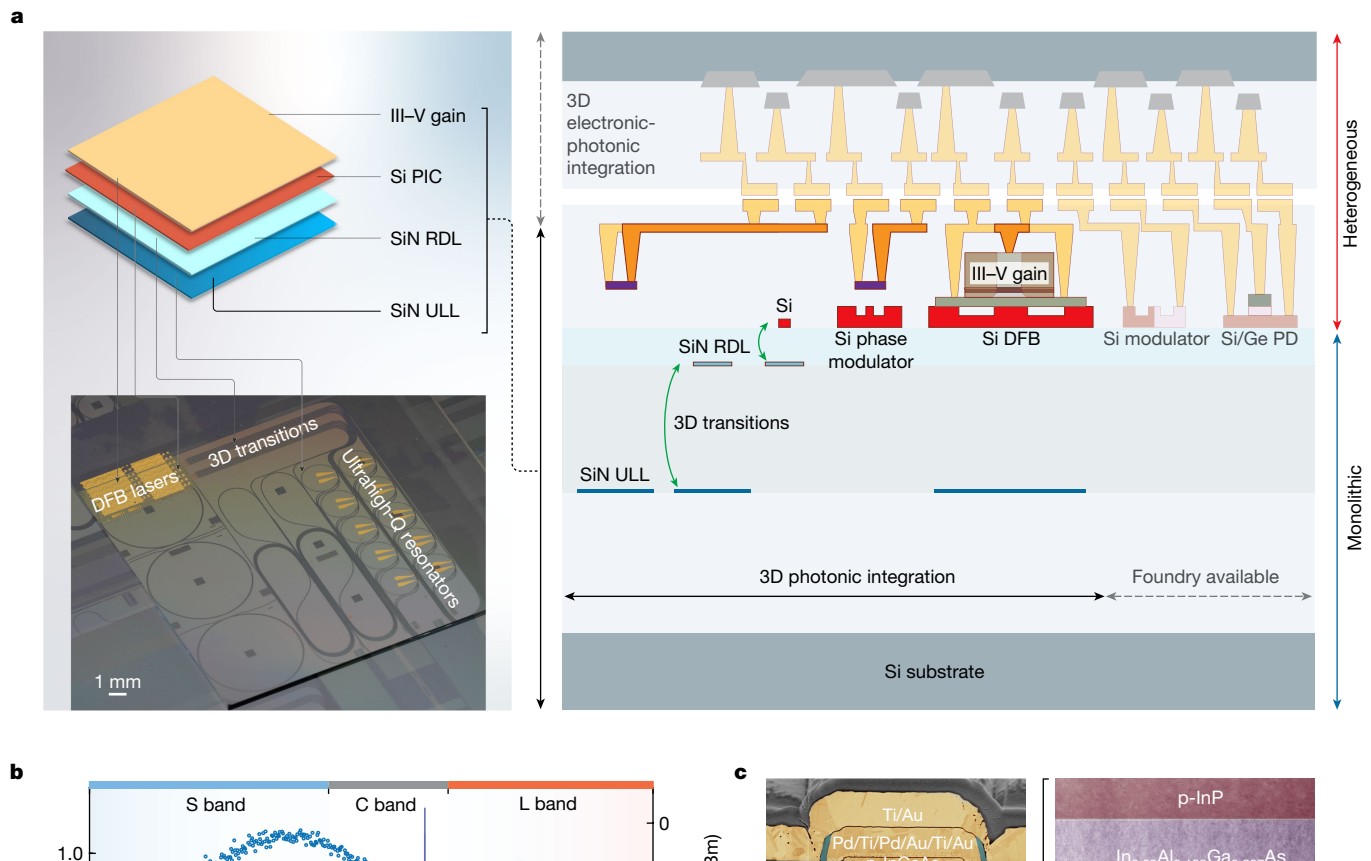

**Fig. 1 | 3D integrated Si PIC chip. a**, Left: concept of 3D photonic integration of functional layers (top) and the corresponding devices on a fabricated 3D PIC (device picture shown in the bottom). This chip is singulated from a fully fabricated 100-mm-diameter wafer. The SiN wafer process is performed on a 200-mm-diameter wafer fabricated in a CMOS foundry, which was later cored into 100-mm-diameter wafers for heterogeneous laser fabrication. Right: the cross-section of the demonstrated 3D PIC in solid colours. We envision that future works will enable additional functionality, such as integration with foundry-available Si modulators and Ge/Si PDs, and 3D electronic–photonic heterogeneous integration, which are shown in transparent colours. Both monolithic and heterogeneous integration processes are employed, in which 3D transitions are critical to the vertical integration of functionality layers. **b**, Measured III–V/Si DFB laser spectrum centred at the telecom C band on the 3D PIC (right axis) and measured ULL SiN waveguide loss (left axis, extracted from the fitted resonator $Q$) across the telecom S, C and L bands on the same 3D PIC. **c**, Left: false-coloured focused ion beam scanning electron microscopy image of the fabricated 3D PIC showing the laser cross-section. Right: transmission electron microscopy image showing the layered InP epitaxial stack after bonding and substrate removal.

## Cavity-mediated feedback sensitivity

In addition to frequency noise, integration with the ultrahigh-$Q$ cavity markedly reduces the feedback sensitivity[38]. This goal has been pursued by many demonstrations, but owing to the difficulty of integrating ultrahigh-$Q$ cavities with lasers, the feedback tolerance is limited such that an isolator is still required to operate in the strong feedback regime (more than −10 dB)[39,40].

In the current SIL configuration with an add-drop ring resonator, the laser output can be taken from both the through and the drop ports (Fig. 3a). The ring resonator itself acts as an intensity filter for both forwards output and backwards reflections. This results in another degree of freedom in controlling the feedback sensitivity by modifying the loading factor of the ring resonator. The dependence on the feedback is characterized using the experimental set-up shown in Fig. 3b. The downstream feedback results in the change of laser coherence for a feedback-sensitive laser. The laser can operate in several different regimes depending on the feedback strength[13]. Stable operation requires the laser to stay in regime I where the laser coherence is maintained. With an increased feedback level, the laser transitions to regime II, where the linewidth is governed by the feedback phase (the length of the external cavity). The critical feedback level at the boundary of regimes I and II ($f_{rI}$) represents the highest feedback level a laser can tolerate to maintain stable operation. After the laser enters regime IV, the laser coherence collapses. Our laser did not enter regime III, in which a significant frequency stabilization owing to external optical

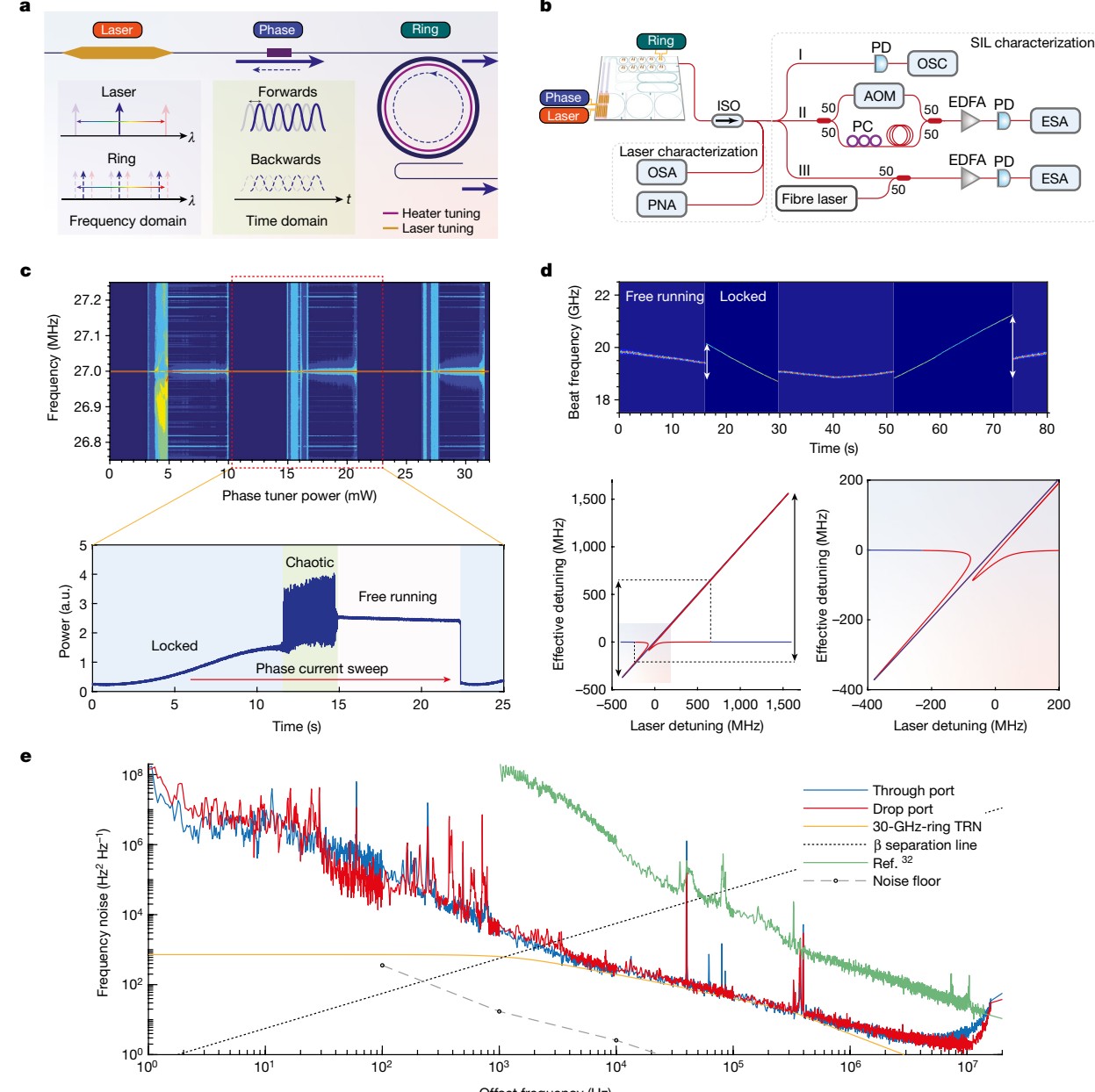

**Fig. 2 | Laser self-injection locking and phase noise. a**, Schematic illustration of the laser self-injection locking, which requires tuning in wavelength and phase to work. There are three knobs used to control the working regimes respectively: laser current, phase heater current and ring heater current. **b**, The experimental set-up to characterize the laser performance and the self-injection locking process. **c**, The dependence of laser self-injection locking on the phase-tuner power. Top: the change in delayed self-heterodyne beat spectrum recorded by an ESA. Bottom: the corresponding power recorded on the oscilloscope of one phase tuning period revealing the locked, chaotic and unlocked states. The acoustic-optic modulator (AOM) used in this experiment has a centre frequency of 27 MHz. **d**, The laser beat frequency with a fibre laser during the ring resonance

blueshift sweep and redshift sweep. The vertical arrows mark the self-injection locking range. The bottom plot is a calculation of asymmetric laser frequency-locking range behaviour without thermal crosstalk for the bidirectional sweep. The blue and red sections of the curve indicate stable and unstable branches, respectively. **e**, The frequency noise of the laser output taken from the through port and drop port of the 30-GHz ring resonator. Comparisons also show the thermorefractive noise (TRN) of the 30-GHz-FSR ring resonator and β separation line. The green curve shows the frequency noise of the SIL laser reported in ref. 32 and the grey dashed curve shows the noise floor of the phase noise analyser (PNA). PC, polarization controller; EDFA, erbium-doped fibre amplifier; ISO, isolator; OSA, optical spectrum analyser; OSC, oscilloscope.

feedback can take place, regardless of the feedback phase. In general, regime III is too narrow to be observed in most semiconductor lasers.

We calculated the critical feedback level as a function of the cavity-loaded $Q$ (Fig. 3c). Subject to different Rayleigh backscattering strengths ($R$), the laser undergoes variable tolerance to downstream reflection. In general, large high-$Q$ feedback (the Rayleigh scattering from the ultrahigh-$Q$ resonator) is beneficial in leading

to high downstream reflection tolerance. This effect saturates at certain loaded $Q$ when the phase response provided by the resonator cannot compensate for larger reflected powers outside the resonator.

To experimentally verify the high feedback tolerance owing to the integrated laser and ultrahigh-$Q$ resonator, we studied the laser coherence dynamics with varied downstream reflection strength, and the

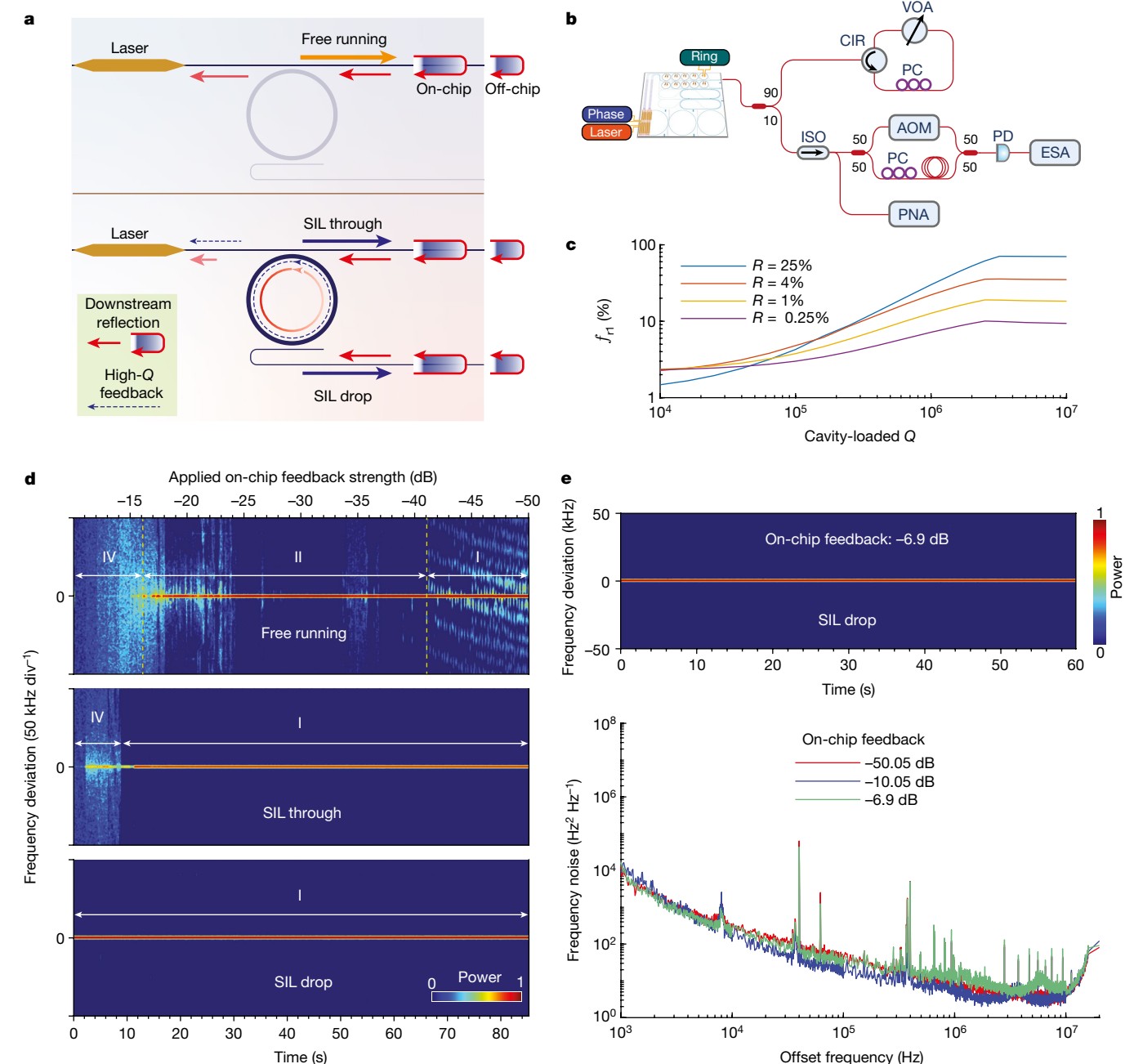

**Fig. 3 | Feedback insensitivity of the SIL laser. a**, Schematic illustration of the feedback influence for the laser working at the free-running state and self-injection locking state. Under self-injection-locking conditions, both the through port and the drop port are characterized. **b**, Experimental set-up for the feedback sensitivity characterization. **c**, Calculation of the dependence of the critical feedback level (the highest tolerable reflection) for regime I boundary ($f_{r1}$) on the cavity-loaded device. The backscatter ($R$) from the ultrahigh-$Q$ resonator also impacts the highest tolerable downstream reflections. **d**, The laser spectral lineshape evolution recorded with an ESA by self-heterodyning with an AOM for the free-running laser state (top), SIL through-port output (middle) and SIL drop-port output (bottom). Different feedback regimes are indicated and details of the regimes are covered in the Supplementary Information. **e**, Frequency noise of the drop port at the SIL condition under different on-chip feedback levels. The top panel shows the recorded laser spectral lineshape evolution under a maximum of −6.9 dB on-chip feedback. CIR, circulator.

results are summarized in Fig. 3d. For the free-running state, the laser enters regime II at an on-chip feedback level of −41 dB. This level of feedback can occur in typical waveguide couplers and splitters. As a result, such feedback sensitivity puts a stringent requirement on the on-chip or off-chip device design if isolators are removed. On the contrary, self-injection locking with a high-$Q$ cavity at both the through and the drop ports sees a clear extended regime I. The critical feedback level for the regime I boundary is increased to −14 dB and more than −10 dB,

respectively. We further increased the downstream on-chip feedback level to the SIL drop port to −6.9 dB (limited by the chip-to-fibre coupling loss) and observed a stable and constant laser linewidth—the same as obtained below −50 dB downstream reflections (Fig. 3e, top). Such 27-dB and over-34-dB improvement in the feedback insensitivity are equivalent to the effective isolation that optical isolators can provide to maintain the laser coherence and thus enable isolator-free, on-chip laser integration with downstream devices that introduce

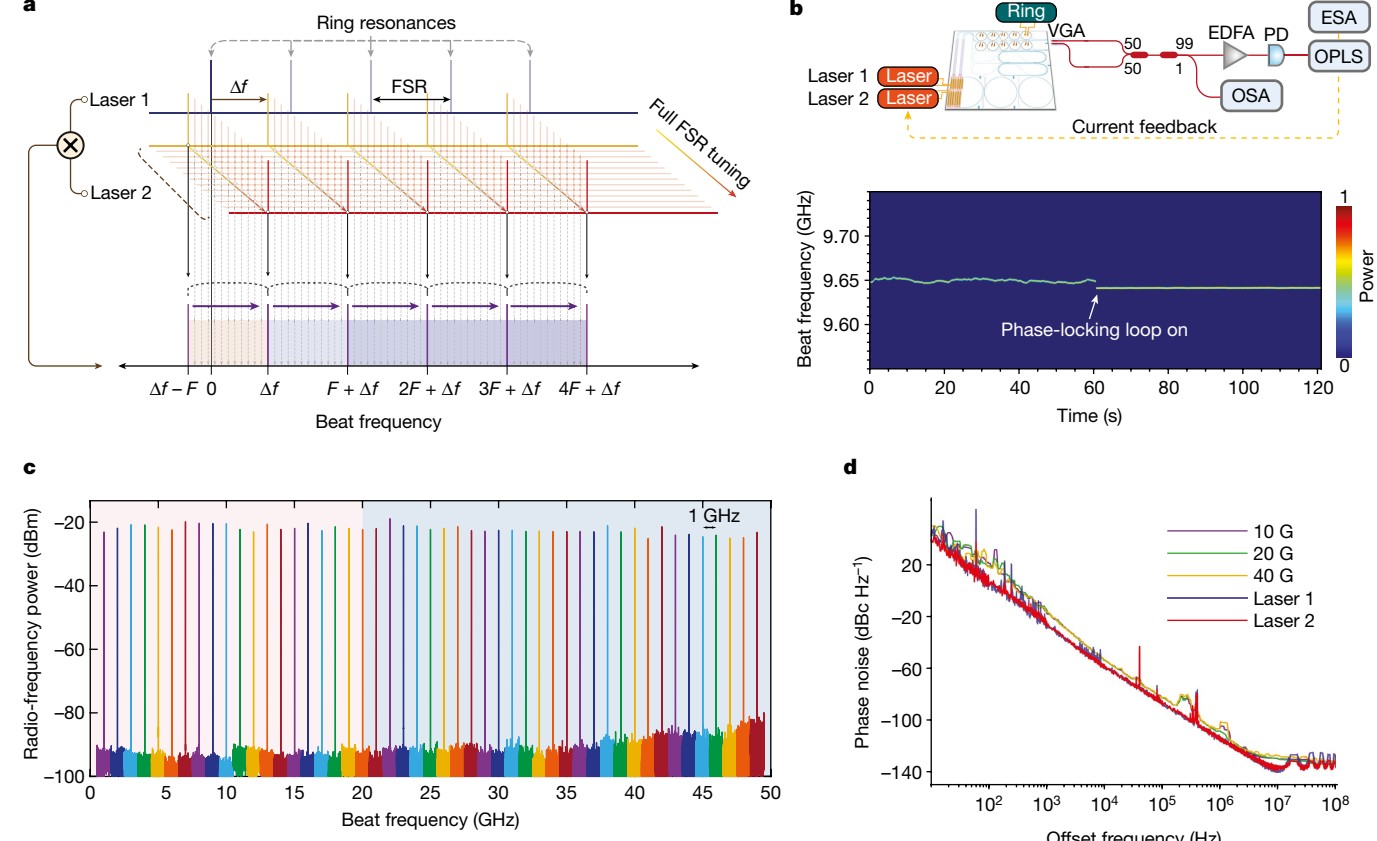

**Fig. 4 | Widely tunable microwave-signal generation. a,** Illustration of the working principle of the widely tunable heterodyne microwave generation based on two SIL lasers. $F$ denotes the FSR and $\Delta f$ is the resonance offset of the two ring resonators without tuning. The shaded-colour regions indicate generated microwave-signal-frequency ranges for a full FSR tuning. **b,** Experimental set-up for the isolator-free, widely tunable heterodyne microwave generation. The bottom inset shows the long-term stability improvement of microwave-frequency generation using a phase-locking loop. **c,** Microwave frequency generated by tuning one SIL laser while keeping the other SIL laser at a fixed wavelength. The highest generated microwave frequency is limited by the PD used in this experiment. **d,** Carrier-frequency-independent phase noise of the generated microwave signal. The heterodyne microwave-signal phase noise is directly determined by the two beating SIL lasers, regardless of the laser frequency separation, that is, the generated microwave frequency. The fitting of the microwave signal is shown in Extended Data Fig. 6. OPLS, offset phase lock servo; VGA, fibre V-groove array.

strong reflection, such as Bragg grating filters, Fabry–Pérot resonant cavities and end-fire coupled components, significantly enriching the complexity of fully on-chip optical systems[41,42]. The frequency noise under different feedback strengths from −50 dB to −6.9 dB is also summarized in Fig. 3e.

## Tunable microwave-frequency generation

The capability of integrating ultralow-noise lasers at the wafer scale opens up the possibility of enabling photonic devices that were impractical to integrate. For instance, microwave-frequency signals can be generated by heterodyne beating two low-noise lasers on a PD with laser frequency offset at the microwave range[43,44]. The generated frequency could be easily tuned by tuning the laser frequency. This schematic is illustrated in Fig. 4a. In such heterodyne beating schemes, the generated microwave-signal phase noise is the sum of the phase noise of the heterodyne beating lasers. Historically, strong semiconductor laser noise prohibited low-noise microwave-frequency synthesis using this scheme. Our demonstrated ultralow-noise lasers provide a route for heterodyne microwave-frequency synthesis on a fast PD directly on a PIC, without additional off-chip linewidth narrowing. The advantage of feedback insensitivity is also critical in direct on-chip microwave synthesis as several components, including the 3-dB couplers and photodiodes, need to follow the lasers, and are potentially strong

sources of on-chip reflection. To verify the feasibility of our lasers for heterodyne microwave synthesis, we performed a tunable microwave synthesis experiment as shown in Fig. 4b. An optical phase-locked loop on driving the laser current can be used to improve long-term stability, as shown in the bottom inset. This stability could be further improved by the chip packaging. The microwave-frequency tuning is achieved by tuning the ring resonances of one ring resonator while keeping the other ring resonance fixed. After laser self-injection locking, the generated microwave-signal frequency is determined by the frequency offset of the two resonances. The generated tunable frequency range is ultimately limited by the PD bandwidth as multiple ring resonances separated by ring FSRs can be used for the locking. For the current lasers, we achieved >3-nm-wavelength separation for the two SIL lasers, corresponding to >375-GHz-heterodyne frequency (Extended Data Fig. 4). The microwave-signal intensity, although affected by the responsivity of the fast PD and the coupling loss in the current off-chip characterization, could be improved by using directly on-chip III–V amplifiers and waveguides and splitters that are fully compatible with our 3D PIC[45].

The generated microwave signals with frequency tuning from 0 to 50 GHz at 1-GHz spacing are summarized in Fig. 4c. The frequency tuning is continuous and determined by the thermal phase-tuner control on the ring resonator. Using a PD with higher bandwidth, we can further extend the generated frequency tuning range. We

characterized the phase noise of the generated microwave signal at different frequencies, as shown in Fig. 4d. It clearly shows that the microwave-signal phase noise is determined by the laser phase noise and is invariant across the microwave carrier frequencies. This unique advantage of heterodyne microwave-signal synthesis is especially promising for widely tunable frequency synthesis without noise penalty at high frequencies, which provides a practical route for low-noise millimetre-wave and terahertz generation. Furthermore, no isolators are used in the experiment, which shows the feedback insensitivity could greatly simplify the system architecture and permit a fully on-chip integrated microwave synthesizer when couplers and PDs are integrated on the same 3D Si PIC[46]. The microwave-generation prototype can be optimized using the same integration platform. For example, locking lasers to the same resonator can take advantage of common noise rejection for several orders-of-magnitude reductions in phase noise[47].

## Discussion and outlook

The demonstrated 3D integration of lasers and ultralow-loss waveguides leverages the advantages of evanescent coupling for vertically spaced photonic functionality layers. This architecture provides a design space for complex on-chip photonic systems without being constrained by in-plane process incompatibility and performance degradation (Extended Data Fig. 7). Many optical devices and systems based on the integration of lasers with optical fibres or separate chips nowadays can be translated onto a Si chip using our demonstrated 3D laser integration with ULL technologies, including Brillouin lasers[48], erbium-doped amplifiers[49], optical gyroscopes[50] and optical frequency synthesizers[51]. Moreover, 3D integration could break the mismatch in the device footprint and density between different waveguide platforms and use the vertical space to improve the device scalability. Our platform could also be used with thick SiN waveguide layers with tight mode confinement for nonlinear applications that require both anomalous dispersion and high-$Q$ cavities[25,26].

The addition of feedback-insensitive ultralow-noise lasers to Si photonics will expand the volume production of Si photonics foundries into applications that remain at small scales. As 3D integration provides an effective solution for multiple functionality layers without compromising performance, more materials and functionalities can be added to the existing integrated platform following certain fabrication process guidelines. These materials include lithium niobate[52], silicon carbide[53], aluminium nitride[54], III–V quantum-dot materials[55] and so on. Our demonstration fuels such explorations and enables new building blocks in integrated photonics. Furthermore, 3D heterogeneous integration with electronics can unite the developments of 3D EICs to enable a 3D E-PIC ecosystem and lay the foundation for a new class of Si chips.

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

## Methods

### 3D mode transitions

Using a series of adiabatic tapers[56], the optical mode is transferred over a vertical distance exceeding 4.8 μm between an InP/Si hybrid mode in the laser active region, to the ULL SiN waveguide layer of the ultrahigh-$Q$ resonator. The mode is transferred first from the InP/Si hybrid waveguide into a Si waveguide and subsequently from the Si waveguide into a SiN RDL. As these InP, Si and SiN layers are fabricated either in contact (in the case of InP and Si) or in close proximity (in the case of the Si and SiN RDL), the optical mode transfers rapidly between them and their tapering lengths are less than 300 μm in total. In particular, the InP to Si rib waveguide transition can be very short (around 25 μm), as InP and Si feature similar refractive indices[45]. The Si rib waveguide with 231-nm etch depth is subsequently tapered to a 200-nm width to transfer the mode to the thin Si waveguide with 269-nm thickness. The thin Si tapers from around 3 μm to 150 nm to match the effective index of the RDL SiN waveguide for efficient Si–SiN power transfer. To span the vertical distance separating the SiN RDL and the ULL SiN layer on which the high-$Q$ resonators reside, the optical mode is gradually evolved from the upper SiN to the lower SiN layer. The RDL SiN and ULL SiN waveguides feature identical core thicknesses of 100 nm, so that their effective indices are readily matched. The RDL SiN waveguide width is thus tapered from 2,800 nm to 200 nm, while simultaneously widening the ULL SiN waveguide width from 200 nm to 2,800 nm, over a distance approaching 1 cm in length. This scheme enables efficient power transfer (<1-dB insertion loss) from the RDL SiN waveguide to the ULL SiN waveguide.

In weakly confined ULL SiN waveguides, the optical mode extends significantly into the silicon dioxide ($SiO_2$) cladding. Previous work[31], in which a ULL SiN waveguide was heterogeneously integrated in close vertical proximity with an InP/Si hybrid waveguide, resulted in a relatively high propagation loss of 0.43 dB cm$^{-1}$. In this work, a 3D layer transition enables the ULL SiN waveguide to be buried deeper within the $SiO_2$ cladding, such that impurities originating from the back-end heterogeneous integration process do not influence the ULL waveguide performance.

To motivate the addition of a SiN RDL in such a 3D layer transition, we compare the performance of a Si-to-SiN waveguide transition with a SiN-to-SiN waveguide transition. The optimal length of an adiabatic coupler transition is given by $L_{opt} \approx \frac{3}{2} \frac{1}{\kappa \cdot \sqrt{\epsilon}}$, in which $\kappa$ represents the coupling coefficient between the waveguides in the coupling region, and $\epsilon = 0.01$ represents the tolerance of power transferred to the undesired, anti-symmetric system mode (that is, loss)[56]. This optimal (minimum) length is calculated as a function of vertical separation in Extended Data Fig. 1a, which demonstrates that a SiN-to-SiN layer results in a more efficient (shorter) transition for vertical separation exceeding 2 μm.

Thus, the inclusion of a SiN RDL beneath the Si waveguide provides improved vertical coupling efficiency, enabling the ULL SiN waveguide to be buried deeper below. The SiN RDL is further motivated by additional performance and fabrication concerns. Efficient power transfer between Si and SiN waveguides requires a very narrow Si width to match the propagation constants of the respective waveguides, as shown in Supplementary Fig. 1b. Such narrow Si waveguides feature significant sidewall-roughness-induced scattering loss, limiting the length of such structures. Furthermore, the combination of narrow width and long length of a Si-to-SiN transition capable of spanning several-micrometre distance would yield a fragile structure that is susceptible to damage during the fabrication process. As such, the close proximity of the SiN RDL to the Si waveguide enables a short Si-to-SiN transition, improving process yield. In this work, the SiN-to-SiN transition length was chosen to be excessively long (approaching 1 cm) to enable flexibility in the choice of vertical separation while retaining transition efficiency.

To experimentally evaluate the achievable transition efficiency from the RDL SiN to the ULL SiN, two such layer transition structures were placed within a racetrack resonator. In contrast to a cut-back approach, in which multiple identical structures are cascaded in series to extract an aggregate insertion loss, a resonator-based measurement technique enables insertion loss of a structure to be measured independently of fibre-to-chip coupling losses, resulting in a more accurate measurement. Previous work has demonstrated this approach to accurately measure insertion losses well below 0.1 dB (ref. 57). The transmission spectrum of the resonator was measured and fitted to extract the internal round-trip loss, as shown in Supplementary Fig. 2. From this measurement, the insertion loss was inferred to be below 0.03 dB per transition. However, the resonator-based test structure was fabricated on a separate wafer that did not undergo any heterogeneous integration process and featured a narrower spacer thickness of roughly 3.5 μm. Thus we conservatively expect the insertion loss of the RDL-to-ULL transition within the heterogeneous laser to be well below 1 dB.

### Device fabrication

Fabrication of the SiN waveguides was performed at Tower Semiconductor, a commercial CMOS foundry, on a 200-mm-diameter Si wafer with 15-μm-thick thermal $SiO_2$. Low-pressure chemical vapour deposition SiN with 100-nm thickness was deposited and patterned using deep ultraviolet (DUV) stepper lithography and reactive ion etching to form the ULL waveguide layer. Tetraethyl orthosilicate-based oxide was deposited on the ULL layer, annealed at 1,150 °C, and underwent chemical mechanical polishing to form an approximately 4-μm-thick spacer layer[35]. To form the RDL waveguide transition, another 100-nm-thick low-pressure chemical vapour deposition SiN was deposited and patterned by the same process. The adiabatic RDL taper was defined and etched on this layer. Additional tetraethyl orthosilicate-based oxide was deposited, annealed and underwent chemical mechanical polishing to leave around 500-nm-thick $SiO_2$ on top of the RDL. The processed 200-mm wafer was then transferred out of the foundry for subsequent processing. The wafer was cored into 100-mm wafers to be compatible with an ASML 248-nm DUV stepper. Diced silicon-on-insulator pieces with 500-nm-thick Si device layer were bonded on the polished $SiO_2$ surface using plasma-activated direct bonding. The Si substrate was removed by mechanical polishing plus deep Si Bosch etching. The buried $SiO_2$ layer was removed by buffered hydrofluoric acid. The fabricated Si/SiN RDL/SiN ULL wafer was then ready for the heterogeneous InP process on Si similar to our previous studies[31]. In general, Si waveguides and tapers were patterned with a DUV stepper whereas the grating was patterned with electron beam lithography with a period of 240 nm. The Si layer underwent several patterned etches with different etch depths. The first etch had a 231-nm etch depth to form Si shallow etched rib waveguides in the InP/Si and phase-tuner sections. Then the Si gratings and thin Si tapers were formed respectively with 269-nm etch depth. Si outgassing channels were patterned later and etched with an etch depth of 500 nm in the area that had no Si waveguides. The Si etch was reactive-ion-etched with a mixed etching gas of $C_4F_8$/$SF_6$ and the etch depth was controlled by an etch monitor Intellemetrics LEP400. After Si processing, InP dies with the layer stack shown in Fig. 1c were bonded on the fabricated Si circuits, with the InP substrate removed by mechanical polishing and 3:1 hydrochloric acid:deionized water. A thin layer of p-type contact-metal Pd/Ge/Pd/Au was formed using a lift-off process. The p-InP mesa was etched using $CH_4$/$H_2$/Ar, with a $SiO_2$ hard mask. The dry etch was monitored using an etch monitor and stopped at the AlInGaAs quantum well (QW) layer. After another round of QW layer lithography, the QW layer was etched using a mixed solution of $H_2O$/$H_2O_2$/$H_3PO_4$ 15/5/1. An n-type InP mesa etch followed the QW etch to complete the mesa definition with the same etching gas $CH_4$/$H_2$/Ar. The excess Si on top of the SiN devices was removed using a $XeF_2$ isotropic gas etch. The entire chip was passivated using

low-temperature deuterated SiO$_2$ (ref. 58) followed by the contact-metal window opening through CF$_4$-based inductively coupled plasma etching. The n-type contact-metal Pd/Ge/Pd/Au and another layer of Ti/Au on top of the p-type contact metal were deposited and formed. Proton implantation was performed to define the current channels. Ti/Pt was deposited as heaters for the phase tuner on Si and resonance tuner on SiN. The chip went through another round of SiO$_2$ deposition and contact via opening. The Ti/Au probe metal was deposited to finish the wafer fabrication. The fabricated 100-mm-diameter 3D PIC wafer was diced and polished to expose the SiN edge couplers for fibre-coupled device characterization. The detailed process flow charts can be found in Extended Data Fig. 7.

## Impurity depth profiling analysis using secondary ion mass spectrometry
To analyse the impurity distribution along the depth direction (depth profiling), a secondary ion mass spectrometry system (also known as ion microprobes, CAMECA IMS 7f) was used to analyse the devices. In the measurement, a raster area of 50 μm × 50 μm was swept with the primary beam (for ionization and sputtering) and secondary ions generated only in the centre area of 20 μm × 20 μm were collected by the instrument filtering aperture to prevent impacts from other layers at the edge of the hole drilled. To obtain conductivity required for secondary ion mass spectrometry, 20 nm of gold was deposited onto device surfaces. Reference devices NIST SRM 610 and 612 (ref. 59) (National Institute of Standards and Technology Standard Reference Materials (NIST SRM)) were used for the calibrations of elementary concentrations. The measurement was implemented in a vacuum level of $3 \times 10^{-9}$ torr. For the positive-ion measurements, O$^-$ ions were the primary beam. For the negative-ion measurements, Cs$^+$ ions were the primary beam, and the electron beam was also engaged to neutralize the sample to avoid charging effects. The results are plotted in Extended Data Fig. 3.

The sample area measured here is a pure waveguide region without the top laser structure but experienced the full back-end-of-line process. The appearance of boron atoms indicates the boundary of the lower thermal oxide cladding layer because the substrate Si wafers are of p-type (resistivity of about 100 Ω cm) to accelerate thick thermal oxidation. The appearances of both Si–N clusters and C–N clusters indicate the thin SiN waveguide layer because nitrogen atoms begin to appear in large amounts. The coincidence of B, Si–N and C–N traces cross-verify each other and gives the SiN waveguide depth position of 5.9 μm.

## Additional laser characterization
The lasers are characterized on a temperature-controlled copper stage with a precision temperature controller (Vescent SLICE-QTC) for device characterization at 20 °C. We screened the lasers before the self-injection-locking characterization, phase-noise measurement and so on. The laser light-current measurement results are shown in Extended Data Fig. 4a, which exhibit an approximately 74-mA laser threshold, influenced by the DFB grating strength. Compared with typical laser light-current behaviours, one difference of such a laser-resonator device is that with the increase of laser gain current, the recorded power would see several dips in the light-current curve when the laser output power is filtered by the ring resonator. The spacing of the resonance dips is determined by the ring resonator FSR (30 GHz in this work) when the laser wavelength is swept across multiple resonances during the gain-current increase. It has to be noted that in this light-current sweep, the laser gain current is stepped at 1 mA so not every resonance can be matched and recorded.

We can thus lock the laser to different resonances by tuning the laser gain current. Besides, the thermal tuning of ring resonances allows the continuous tuning of the SIL laser wavelengths across the DFB laser wavelength. This capability is critical in microwave generation

as microwave frequency can be synthesized precisely based on the laser gain and ring resonance controls. We lock two SIL lasers at two resonances with over 3-nm-wavelength space and the laser spectra are shown in Extended Data Fig. 4b. This wavelength separation promises >375-GHz millimetre-wave generation if a fast PD is available. More importantly, the phase noise will be the same as low carrier frequencies as it is determined by the laser phase noise.

## Laser self-injection locking
By tuning the laser wavelength to a resonance from the ring, the backscattered light from the ring locks the laser wavelength to the resonance provided that the phase of the backscattered light arriving at the laser is an integer multiple of 2π of the forwards laser output phase. In other words, the wavelengths of the laser and the resonance are matched in the frequency domain whereas the phases of the laser and the backscattered light are matched in the time domain, as shown in Fig. 2a. Matching the wavelengths is performed by tuning either the laser gain current or the ring heater current, whereas matching the phases is done by tuning the phase-tuner current. Both laser gain current and phase-tuner current are driven with low-noise laser current sources (ILX Lightwave LDX-3620) to ensure stable and low-noise operation. Detection of the self-injection-locking state is assured by observing not only the decrease in the output power from the ring when the laser wavelength hits the resonance but also the decrease in the linewidth of the self-heterodyne beat as the self-injection locking takes place, as shown in Fig. 2b. The self-heterodyne interferometer set-up consists basically of a Mach–Zehnder interferometer (made from two 3-dB couplers) with a polarization controller and a short delay line in one of its arms and a fibre-coupled acoustic-optic modulator (Gooch & Housego 27 MHz) in the other arm, as shown in Fig. 2b. The beat frequency from the self-heterodyne interferometer is detected using a PD (Newport 1811) before it is sent to an electrical spectrum analyser (ESA) (Rohde & Schwarz FSWP). The phase-tuner current is roughly adjusted during the self-injection-locking process to allow the locking to occur and finely tuned afterwards to ensure stable self-injection locking. It is worth mentioning here that the self-injection-locking state can last for hours without even packaging the laser chip. This can be attributed to the integration of the laser and the resonator on the same chip, which reduces the phase fluctuation between the laser and the backscattered light from the ring.

## SIL laser characterization
The dynamics of the phase-tuner influence on self-injection locking is investigated by sweeping its applied electrical power (Keithley 2604B) over three 2π injection-locking periods while recording the ESA spectrogram of the detected self-heterodyne beat of the SIL laser (Fig. 2c, top). The spectrogram of the injection-locking periods, which is depicted in Fig. 2c, demonstrates stable SIL periods (dark blue regions) followed by chaotic regions (light blue) and then unlocked regions. The laser power is also detected on an oscilloscope (Tektronix MSO64B) during the phase tuning over only one period, which clearly shows the mentioned behaviour (Fig. 2c, bottom). Another important parameter is the frequency range at which the self-injection locking persists. It can be obtained by either sweeping the laser frequency over the ring resonance or sweeping the ring resonance over the laser. We selected the second scheme by sweeping the current of the ring heater using a triangle signal applied to the current source (Keithley 2604B). To detect the change in laser linewidth during sweeping, a beat is made using a 3-dB coupler between the SIL laser and a narrow-linewidth fibre laser. The beat is optically amplified with an erbium-doped fibre amplifier (Amonics AEDFA-IL-18-B-FA) and sent to the fast PD (Finisar HPDV2120R) that is connected to the ESA, as shown in the lower branch of Fig. 2b. The recorded spectrogram during the resonance sweep is shown in Fig. 2d. The laser frequency noise and resultant fundamental linewidth are taken from a commercial laser phase noise analyser

(OEwaves OE4000) that internally performs averaging over the measured phase noise. We have not observed significant differences in our noise spectrum between 1 kHz and 1 MHz for several measurement runs, which are very stable, and believe that the noise spectrum in this range is dominated by the thermorefractive noise of the resonator by comparing with the simulation results (Fig. 2e). As a comparison, the delayed self-heterodyne set-up uses two PDs to receive the heterodyne beat[60], which has been previously used for ultralow-noise laser linewidth characterizations[36] and allows for a more detailed analysis of statistical measurement errors.

## Laser feedback sensitivity measurement

Figure 3b schematically depicts the experimental configurations for analysing the laser feedback sensitivity. The coupled laser emission is sent to a 90/10 fibre beamsplitter, after which 90% of the coupled power will be used for external optical feedback. The feedback loop consists of an 8-m-long single-mode fibre, a three-port optical circulator, a polarization controller and a variable optical attenuator (VOA) that allows for an attenuation ranging from 0 to 40 dB (EXFO MOA-3800). It should be noted that the laser feedback sensitivity also depends on the polarization of the reflected field, which must be adjusted to maximize the feedback influence before running the analysis. The remaining 10% of laser output is utilized for the feedback sensitivity characterization. After passing through an optical isolator, it is transferred either to a phase noise analyser (OEwaves OE4000) for frequency-noise characterization or a delayed self-heterodyne set-up for the electrical spectrum-based laser coherence check.

In this study, the feedback strength is determined by the reflected power ($P_{refl}$) and the output power ($P_{out}$) through the following relationship:

$$\eta_F = \frac{P_{refl}}{P_{out}} \tag{1}$$

All losses from the feedback loop should be considered to calculate the reflected power, thus the feedback strength. After optimizing the set-up, the fibre-chip coupling loss is −3 dB (round-trip coupling loss is −6 dB), the total losses from the 90% beamsplitter, the optical circulator, the insertion loss of the VOA, the polarization controller and the fibre is −4.05 dB. The feedback strength that accounts for the attenuation of VOA can thus be tuned from −10.05 dB to −50.05 dB.

To further reduce the loss from set-up and thus maximize the feedback strength as large as possible, we use a 100% fibre back-reflector (BKR, Thorlabs) to replace the configurations after the 90% beamsplitter port. The loss from the feedback loop is then reduced to −0.9 dB, and the maximum feedback strength is −6.9 dB.

## Microwave-signal generation

Two lasers are SIL to two 30-GHz-FSR ring resonators that are shifted in frequency by 10 GHz without tuning on the ring resonance. Although each ring resonator can be tuned over 30 GHz by applying around 0.5 W of electrical power to the heater on the ring, we used only one ring heater for tuning. The tuning range for the first FSR is −10 GHz to 20 GHz, whereas the next full FSR tuning leads to 20 GHz to 50 GHz tuning by locking the second laser to the next resonance and hence covering the full 50 GHz. As shown in Fig. 4b, the two lasers' outputs are collected from the chip by the fibre V-groove array and sent to a 3-dB fibre coupler, then the erbium-doped fibre amplifier before beating on a fast PD (Finisar HPDV2120R) connected to an ESA. A small portion of the laser output (1%) is sent to an optical spectrum analyser

(Yokogawa AQ6370C) for monitoring if the lasers are on the intended resonance. Although our chip could be used to generate any arbitrary microwave frequency over 50 GHz, it is used here to generate microwave frequencies at steps of 1 GHz over the full 50 GHz for demonstration (Fig. 4c). An offset phase-locking servo circuit (Vescent D2-135) is used to improve the long-term stability of the generated microwave signals for frequencies up to 10 GHz, by locking the phase of one of the lasers to the other one. The feedback signal from the servo control box is sent to one of the laser's ring heaters to lock its phase to the phase of the second laser. Stable microwave signals are thus generated with low-phase-noise characteristics of the SIL lasers and the Voigt fitting is plotted in Extended Data Fig. 6.

## Data availability

The data used to produce the plots within this paper are available at https://doi.org/10.5281/zenodo.7894620 (ref. 61).

## Code availability

The code used to produce the plots within this paper is available at https://doi.org/10.5281/zenodo.7894620 (ref. 61).

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

**Acknowledgements** This work is supported by the Defense Advanced Research Projects Agency (DARPA) MTO GRYPHON (HR0011-22-2-0009) and LUMOS (HR0011-20-2-0044) programmes. We thank A. Netherton, M. Li, F. Quinlan and G. Keeler for discussions. O.T. acknowledges support from the Fulbright Scholar Program. A portion of this work was performed in the UCSB Nanofabrication Facility, an open-access laboratory.

**Author contributions** C.X. and W.J. led the 3D PIC device design. C.X. led the heterogeneous integration and device characterization. W.J. designed the SiN devices and 3D couplers. W.J., A.F. and M.P. developed the bilayer SiN fabrication process and handled the SiN wafer processing. C.X. and J.P. fabricated the 3D PIC device with assistance from W.J. O.T. and C.X. characterized and gathered the experimental data from the device, including laser noise, locking ranges, phase tuning and microwave generation, with contributions from J.G., B.D., T.J.M. and Q.-X.J. B.D., O.T. and C.X. performed the feedback sensitivity measurement. H.W. provided theoretical calculations and analysis on the locking dynamics and feedback sensitivity. L.W. performed secondary ion mass spectrometry concentration analysis of the device. E.H. took the focused ion beam scanning electron microscopy and transmission electron microscopy images of the device. C.X. wrote the paper with input from W.J., H.W., B.D., O.T. and J.G. All authors commented on and edited the paper. K.J.V. and J.E.B. supervised the project.

**Competing interests** J.E.B. is a co-founder and shareholder of Nexus Photonics and Quintessent, start-ups in silicon photonics.

**Additional information**
**Correspondence and requests for materials** should be addressed to Chao Xiang or John E. Bowers.

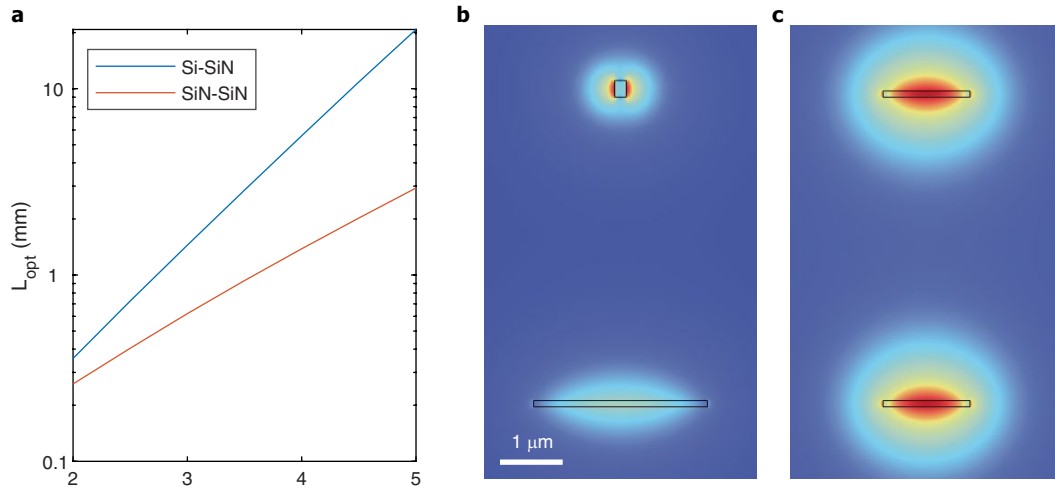

**Extended Data Fig. 1 | 3D layer transition design of Si-SiN and SiN-SiN with large vertical separation. a**. We compare the minimum transition length for a SiN-to-SiN transition and a Si-to-SiN transition over a vertical distance spanning 2 μm to 5 μm. The SiN-to-SiN transition transfers power more efficiently for vertical separations exceeding 2 μm, enabling a correspondingly shorter transition length. **b**. Example of the symmetric supermode of a Si-to-SiN vertical transition. **c**. Example of symmetric supermode in a SiN-to-SiN vertical transition.

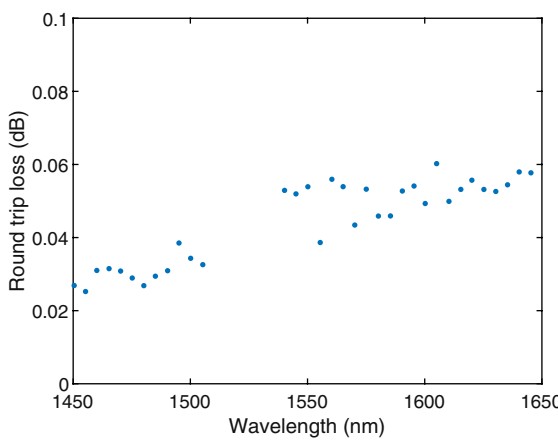

**Extended Data Fig. 2 | The round-trip loss of two RDL-ULL layer transitions.**
Formed into a resonator, the transition loss was determined by fitting the
resonator transmission spectrum, at various wavelengths within the C-band.
As the total round-trip loss is below 0.06 dB, the insertion loss of a single
transition is inferred to be below 0.03 dB (half of the round-trip loss).

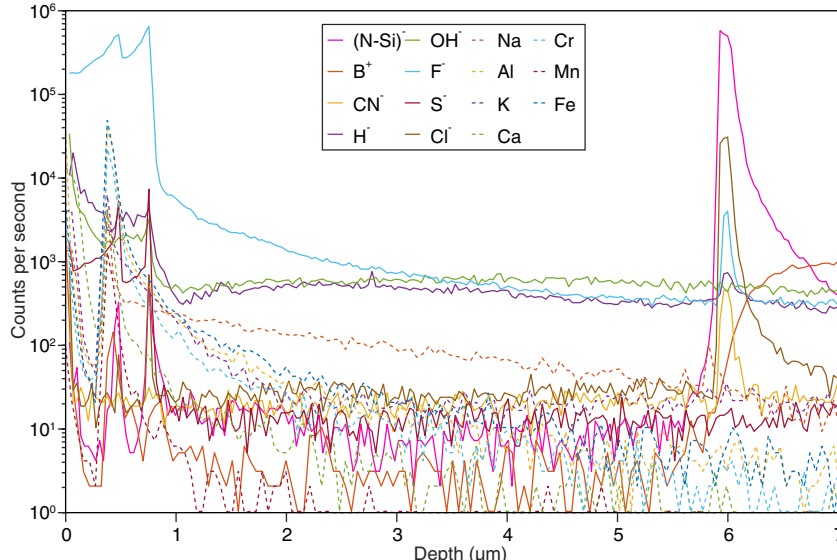

**Extended Data Fig. 3 | Impurity depth profiling using Secondary Ion Mass Spectrometry (SIMS).** The impurity profiles show the effective isolation of loss origins from SiN ULL thanks to the thick upper cladding enabled by 3D integration.

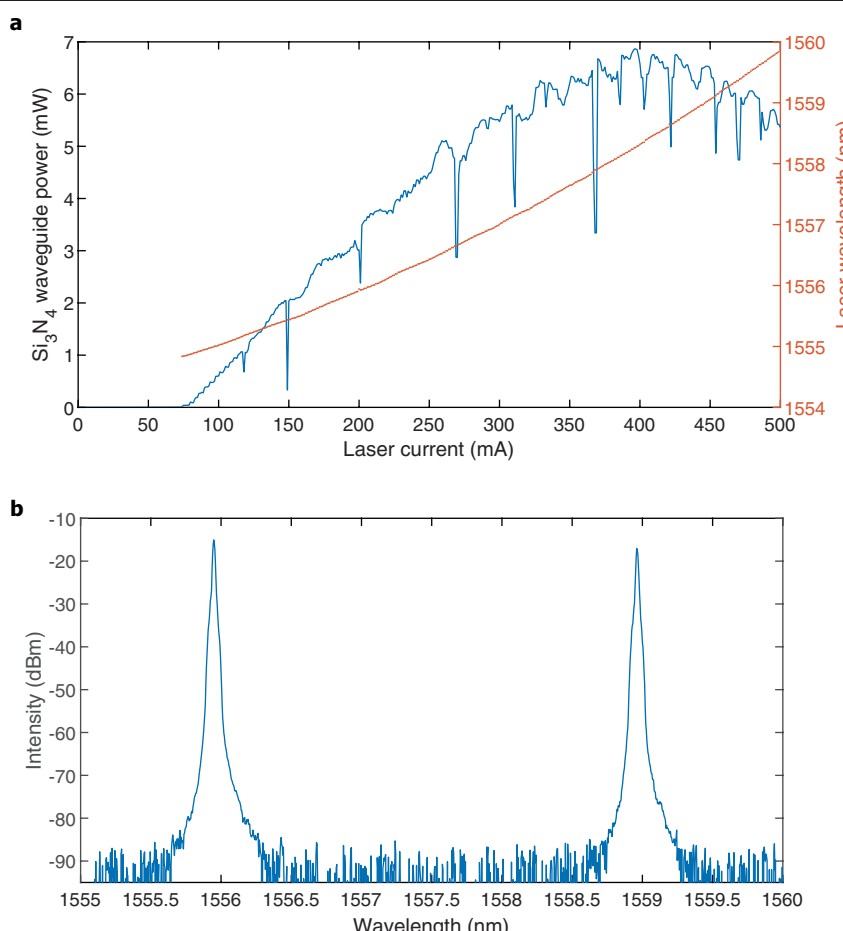

**Extended Data Fig. 4 | Additional laser characterization. a**. Light-current and lasing wavelength measurement of the DFB laser taken from the SiN waveguide facet coupler. Multiple dips in output power during the current sweep exist due to the intensity filtering of the 30-GHz-FSR ring resonator. **b**. Laser spectra showing > 3 nm wavelength separation of the two SIL lasers. The output of the two lasers can be used for > 375 GHz low-noise millimeter-wave heterodyne signal generation.

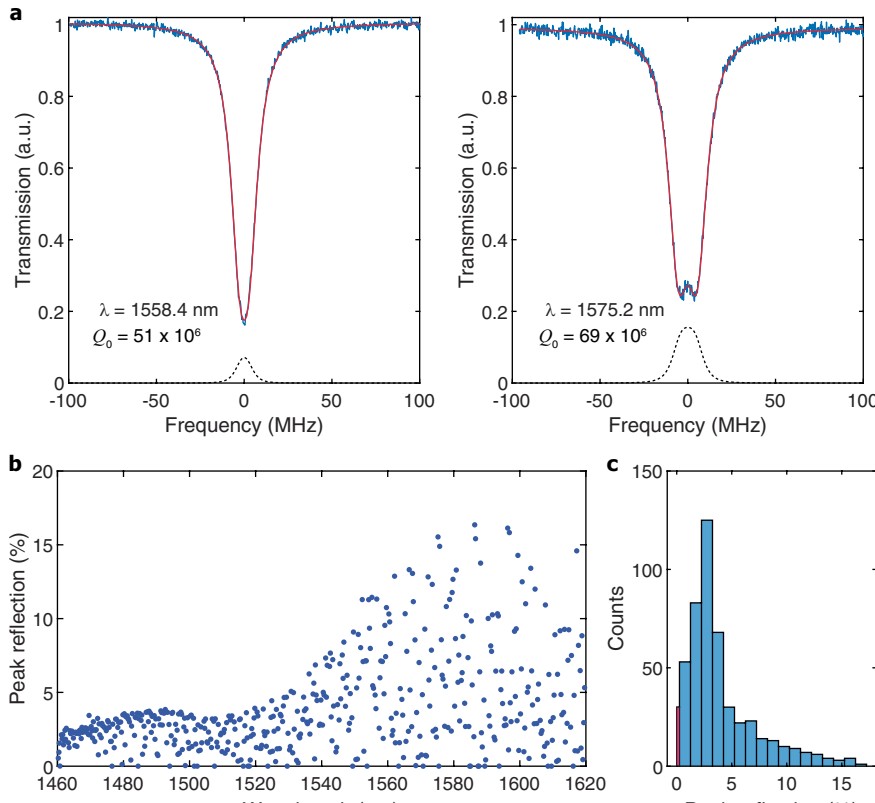

**Extended Data Fig. 5 | Resonator quality factor fitting and reflection extraction. a**. Measured power transmission spectra of two resonances of an independent ring resonator are shown (blue traces). The resonator originates from an adjacent die of the same wafer as the heterogeneous laser devices, and underwent identical processing. The spectra are measured by frequency-scanning a bench-top tunable laser, with the transmitted power collected by a fiber-coupled photodiode, and with the laser scanning rate simultaneously calibrated by a fiber Mach-Zehnder interferometer with 10 m arm imbalance. A resonator model of the transmission is fitted (red curves) to the measured lineshape, from which the resonator parameters, including propagation loss, coupling, and back-scattered reflection, are inferred. An estimate of the resonator back-scattering is extracted from the same model and plotted (dashed black curves). The left-panel resonance does not exhibit an obvious

splitting in the lineshape, whereas the right-panel resonance does. However, both resonances are predicted to provide a substantial back-scattering. **b**. The peak reflection of each resonance wavelength across the S+C+L bands is extracted from the same data set as Fig. 1b. **c.** Histogram statistics of the extracted peak reflection for 505 resonances. The red-colored marked area shows the resonance counts (30) with peak reflection below 0.25%. The most probable peak reflection is around 3%. While the reflection coefficient is variable between resonances, most resonances exhibit sufficient reflection for self-injection locking. Considering the capability of resonance tuning, the resonance reflection would not affect the self-injection-locked laser yield. More control on the back reflection strength could be implemented by leveraging grating structures within the ring resonator[62].

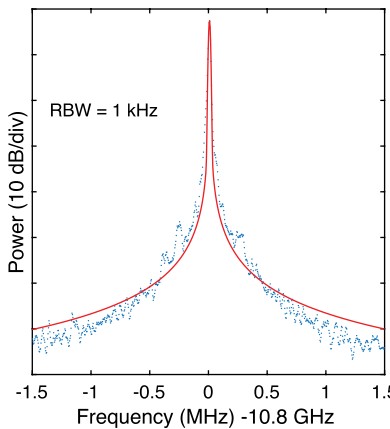

**Extended Data Fig. 6 | Microwave signal spectrum and fitting.** The spectrum of the generated microwave signal with a center frequency of 10.8 GHz is plotted in dotted blue, with Voigt fitting (in red) applied to extract the Gaussian linewidth of 15 kHz.

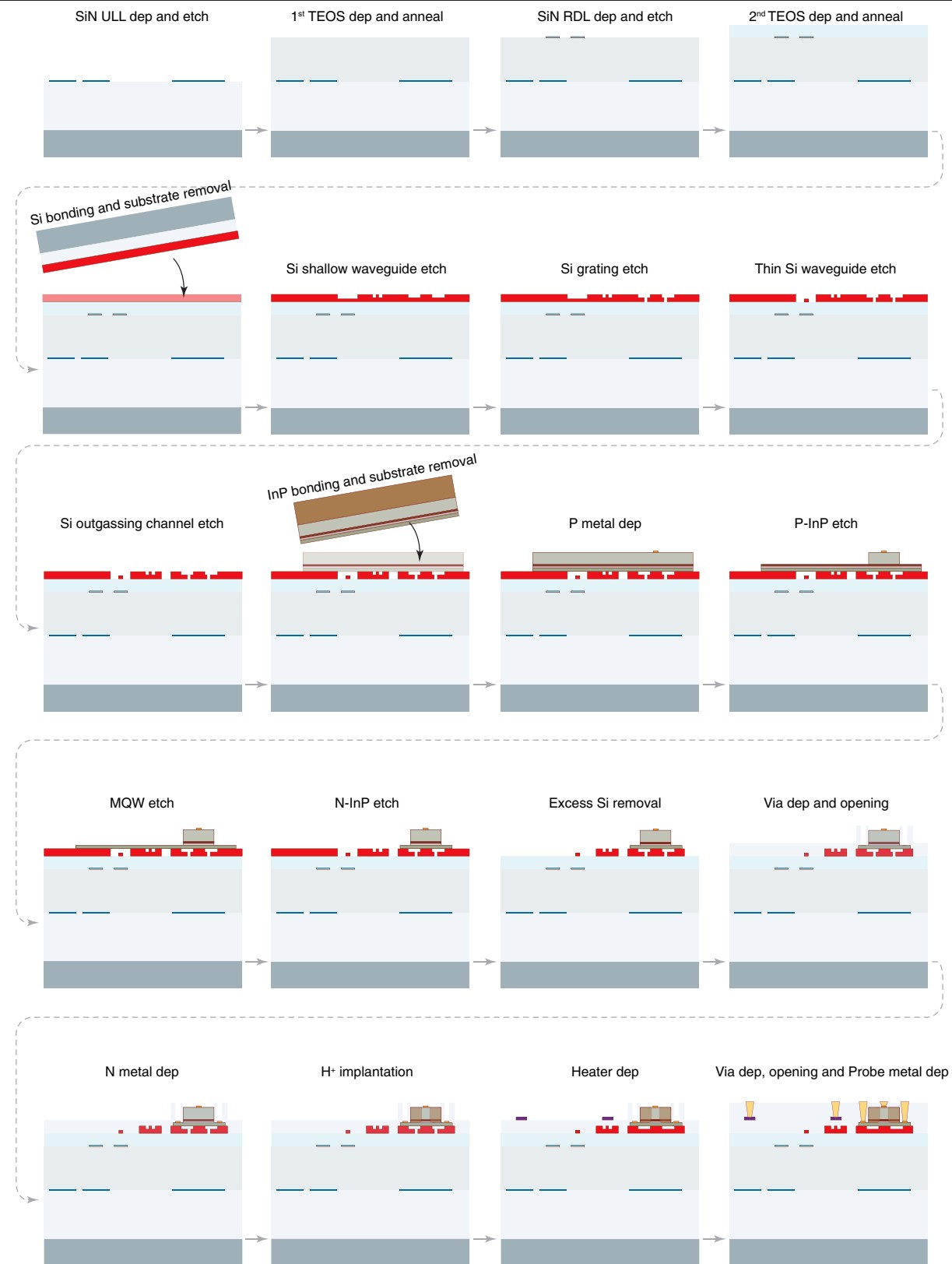

**Extended Data Fig. 7 | Detailed process flow charts.** The details of the respective layers can be found in Fig. 1. The difference in the III-V/Si section between the last two charts and the previous charts are illustrations of the etched Si grating.