## [Peer Review File · Nature]

Manuscript Title: 3D integration enables ultra-low-noise isolator-free lasers in Si photonics

Reviewer Comments & Author Rebuttals

Reviewer Reports on the Initial Version:

Referees' comments:

Referee #1 (Remarks to the Author):

The authors demonstrate a heterogeneously integrated low-noise, feedback-insensitive III-V on silicon laser. This is achieved through 3D integration of a low-loss silicon nitride waveguide layer within the stack that can remain isolated from the active gain materials leading to the high-Q resonator cavity that is needed for high-performance self-injection locked lasers. They further demonstrate its practical use through microwave frequency generation up to 50 GHz with low phase noise through heterodyne beating of two such lasers on a single chip.

Laser integration on the silicon photonics platform has been a longstanding challenge in the field. As the authors claim, heterogeneous integration of III-V materials on silicon seem to be the most viable route. It is indeed true that the instability of lasers due to feedback from downstream circuit elements has been a significant challenge and despite years of research on integrated isolators, there is not a satisfying solution yet. Most require complicated fabrication processes. Building in this robustness to back-reflections and inherent low-noise properties into the lasers are then absolutely essential for future silicon photonic systems. It should be noted though that integrated optical isolators would still be useful as there may be other feedback-sensitive elements in addition to the laser.

The manuscript does an excellent job demonstrating the noise and feedback-insensitive properties of these lasers. This has been especially challenging to demonstrate in self-injection locked lasers because most demonstrations have used hybrid integration schemes where the lasing chip is butt-coupled to the integrated resonator chip without proper packaging leading to alignment instability. However, the performance (linewidth, noise, feedback insensitivity) and fabrication complexity of the demonstrated laser has not been put in proper context relative to the most recent previous demonstrations many of which are heterogeneously integrated and many of which are works by the same authors. This comparison should be very explicit and quantitative. A few references to consider for comparison: Cuyvers, *Lasers & Photonics Reviews*, 2021; Xiang, *Nature Communications*, 2021; Duan, *IEEE PTL*, 2019; Guo, *Science Advances*, 2022; Zhang, *IEEE JLT*, 2020.

The heterogeneous integration of this high-Q silicon nitride cavity of 50 million to the active gain material is an impressive demonstration in itself. To achieve this, they have used a low confinement silicon nitride waveguide which allows for less interaction with fabricated surfaces of the waveguide. This platform is excellent for pushing the Q very high, even above 100 million, as shown in their previous works. However, it also requires larger adiabatic mode transitions, larger bends, and a 15 micron oxide underneath. The fabrication process, mode transitions throughout the device, and foundry compatibility of these processes is not discussed at all, so it is difficult to judge their compatibility and this is a major claim of the work. It is also not clear in the manuscript what is the real impact on photonic applications of going beyond 10 million Q on the laser performance. It seems at least the feedback-insensitivity saturates after a Q of about 2 million. The linewidth and noise do have a strong proportional dependence on the Q, but it would be important to note which applications require such low noise lasers as for most application ~kHz is acceptable.

The microwave frequency demonstration was quite impressive as the low-noise of the currently demonstrated lasers, despite not being locked to the same cavity, demonstrate phase noise

properties much closer to previous integrated frequency comb demonstrations. However, this demonstration is missing some details such as the measurement of the linewidth of the new microwave frequency generation and exact numbers for the phase noise to compare it with state-of-the-art integrated microwave generation. The claim is that now that the lasers are at such low noise levels they can generate microwave frequencies with low noise levels as well, and this does seem true in the demonstration. References to consider explicitly comparing with: Hulme, Optics Express, 2017; Liu, Nature Photonics, 2020.

The demonstration in the manuscript is impressive, but is missing important details which would place the work in proper context and show whether this is truly a major achievement over the most recent demonstrations. For that reason, I recommend the authors must address the concerns above and the following additional points in a future revision to be accepted:

1. There should be a measurement of the quality factor of the ring resonator as one of the major claims is that the noise and feedback-insensitivity is strongly dependent on this.
2. How are the losses measured in Fig. 1?
3. How was the linewidth of the laser measured? The measurement and details are missing. This should have measurement error statistics.
4. What would be the theoretical limit of the noise and linewidth based on Q's that can be achieved on this platform.
5. It is difficult to understand the full mode transition of the 3D vertical transition from the active gain material of the laser to the high-Q silicon nitride layer. All the mode transitions and fabrication processes should be included in the supplementary.
6. The high-Q shown here can also be achieved in a high confinement sin platform that does not create such large mode sizes, are there advantages or disadvantages that pertain to the noise and feedback-sensitivity characteristics of the laser in high confinement vs. low confinement cavities?
7. A measurement of the linewidth of the microwave frequency generated is missing and a calculation or description of how this is quantitatively related to the laser properties.
8. How is the R value, feedback fraction, calculated? Can this be compared with optical isolator specifications that would be needed to achieve the same performance?
9. How does the feedback insensitivity of the lasers affect the microwave frequency generation?
10. It is mentioned that increasing the resonator cavity radius can reduce the thermorefractive noise. Would this imply a fundamental tradeoff between the mode-hop free tunability of the laser and noise?
11. Fig. 2e should include noise floor of the instrument as well.
12. Last main text figure has a letter labeling typo.

Referee #2 (Remarks to the Author):

The authors demonstrate an isolator-free self injection locked laser platform based on 3D integration. They demonstrate 3D integration of lasers with low loss waveguides, self injection locking of the lasers, isolator free operation, and tunable microwave frequency generation. The work is very well written and clearly presented. The manuscript is technically sound and aside from a minor issue with figure 4 is excellent. My main concern is that, in my opinion, the manuscript seems too specialized, as presented, for Nature. The manuscript goes in depth into the injection locking and the noise measurements. Additionally, the authors have presented significant results in prior publications on this topic. The main advance seems to be bringing these prior innovations together into one device.

Minor Concern:

- In figure 4, there appears to be a mismatch between the text, the figure, and the figure caption. The figure only includes parts a through d, while the caption mentions part e. Based on the figure caption it seems to me that the authors meant to label the beat frequency vs time plot as part c.

This should be corrected.

Referee #3 (Remarks to the Author):

The work reports a multi-layer approach for the heterogeneous integration of ultra-low-linewidth lasers in silicon photonics. Two layers of ultra-low-loss silicon nitride are stacked on top of each other, followed by a silicon photonic PIC and III-V gain chiplet. The lasers are implemented by injection locking a distributed feedback laser to a high-Q optical microresonator cavity. The DFB laser is implemented in the Si-III-V stack, taking advantage of the active gain provided by the InP material, and the ultra-high-Q cavity is implemented in the bottom SiN layer.

Although similar 3D integration approaches have been developed in recent years, as the authors recognize in the reference list, this is the first time to my knowledge that the SiN layer displays losses in the dB/m range. This accomplishment is crucial for attaining a sufficiently high-quality factor in the resonator used for the external cavity and reach an integrated laser with excellent coherence performance. Critical to this is the development of the intermediate SiN layer, which helps to bury the ULL layer at the bottom, far from the semiconductor stack, and minimize impurities that would otherwise degrade the loss performance. The redistribution layer also assists in the realization of presumably efficient transitions between the distinct layers and materials.

Another important aspect of the study is the resilience of the laser architecture to spurious reflections. The authors do a remarkable analysis of the laser dynamics and show that with sufficiently high Q, the laser could withstand large reflections without affecting the injection locking to the cavity nor the purity (coherence). The significance of the architecture, as the authors recognize, is that the laser avoids the use of optical isolators, which are complicated to manufacture and typically require materials that are currently not compatible with the CMOS processing lines.

The content of the manuscript is extremely well organized. The figures are clear and succinct, and I find an excellent balance between technical detail, forward-looking discussions and rigor.

In my opinion, the achievement of heterogeneous integration of ultra-low-linewidth lasers in silicon photonics deserves the visibility that a publication in Nature provides.

I only have a couple of remarks that I warmly encourage the authors to consider for clarifications:

1. The introduction gives at instances the false impression that this is the first time that a 3D integration approach in silicon photonics is reported including active devices. However, references 32, 34 and 35, just to take some from the reference list, have developed similar multilayer integration techniques that include active components. I believe the manuscript achieves for the first time a multilayer integration approach between actives and passives that preserves the ultralow losses in the SiN layer, but this message is not clearly conveyed in the introduction, hence decreasing the value/merit of the previous studies in the field.

2. The injection locking technique is now well established and relies on the Rayleigh scattering in the high-Q microresonator coupling forward and backward modes. This process is non-controllable, raising concerns over how repeatable the design is and what is ultimately the fabrication yield. The analysis in Figure 3c indicates that in the high Q regime, there is a large range of scattering coefficients that would result in a laser design that is resilient to reflections $< 10\%$, but still, how likely is it to get a scattering coefficient with $R > 0.25\%$? Does this happen for all wavelengths, i.e., do all resonant modes display a split resonance profile? What are the implications on the laser yield?

3. The fabrication details are quite slim. For example, is CMP only implemented after the deposition of the second layer? What is the homogeneity and roughness of the films? What strategy is used to properly align the masks? Is annealing implemented at some step to achieve ultralow losses? If so, under what conditions?

4. The analysis of the 3D transitions would benefit from clearer illustrations. Also, are the transition losses stated in the manuscript measured or calculated? I appreciate it is challenging to conduct measurements of the crosstalk and losses, especially when the transitions are very efficient, but it would be helpful to share the data with the community if available.

Author Rebuttals to Initial Comments:

We appreciate the careful review by the three reviewers and have modified the manuscript in accordance with their suggestions. We are grateful for the positive evaluation from all the reviewers about our results and the suggestions to include more technical details for a stronger paper. In the revised manuscript, according to the reviewers' suggestions, **we have added new schematics and data in the Supplementary Information (SI), Extended Data Figures and more explanations in the main manuscript.** In the following, we present a point-by-point reply (in blue.) to the reviewers' comments (in black), as well as the action taken (in red).

Referees' comments:

Referee #1 (Remarks to the Author):

The authors demonstrate a heterogeneously integrated low-noise, feedback-insensitive III-V on silicon laser. This is achieved through 3D integration of a low-loss silicon nitride waveguide layer within the stack that can remain isolated from the active gain materials leading to the high-Q resonator cavity that is needed for high-performance self-injection locked lasers. They further demonstrate its practical use through microwave frequency generation up to 50 GHz with low phase noise through heterodyne beating of two such lasers on a single chip.

Laser integration on the silicon photonics platform has been a longstanding challenge in the field. As the authors claim, heterogeneous integration of III-V materials on silicon seem to be the most viable route. It is indeed true that the instability of lasers due to feedback from downstream circuit elements has been a significant challenge and despite years of research on integrated isolators, there is not a satisfying solution yet. Most require complicated fabrication processes. Building in this robustness to back-reflections and inherent low-noise properties into the lasers are then absolutely essential for future silicon photonic systems. It should be noted though that integrated optical isolators would still be useful as there may be other feedback-sensitive elements in addition to the laser.

Our reply:

We thank Reviewer #1 for the positive evaluation of our work. We agree there are other feedback-sensitive elements that might also need isolators. To avoid misunderstanding, we included a description that the 'isolator-free' operation is meant for the integrated laser in both revised title and abstract text.

Our action taken:

- We changed the title to be: "3D integration enables ultra-low-noise isolator-free lasers in Si photonics"
- We rephrased in the abstract: "leveraging three-dimensional (3D) integration that results in ultra-low-noise lasers with isolator-free operation for silicon photonics".

The manuscript does an excellent job demonstrating the noise and feedback-insensitive properties of these lasers. This has been especially challenging to demonstrate in self-injection locked lasers because most demonstrations have used hybrid integration schemes where the lasing chip is butt-coupled to the integrated resonator chip without proper packaging leading to alignment instability. However, the performance (linewidth, noise, feedback insensitivity) and fabrication complexity of the demonstrated laser has not been put in proper context relative to the most recent previous demonstrations many of which are heterogeneously integrated and many of which are works by the same authors. This comparison should be very explicit and quantitative. A few references to consider for comparison: Cuyvers, *Lasers & Photonics Reviews*, 2021; Xiang, *Nature Communications*, 2021; Duan, *IEEE PTL*, 2019; Guo, *Science Advances*, 2022; Zhang, *IEEE JLT*, 2020.

Our reply:

We thank the reviewer for the acknowledgement of the difficulty in realizing heterogeneous laser integration with ultra-high-Q resonators. As the reviewer pointed out, this integration task so far has been dominantly achieved by hybrid butt coupling, which is however, not a highly-scalable and highly-robust approach to achieve dense photonic integrated circuits (PICs) and ultimate stability. We want to emphasize that, in addition to scalability and stability, the device performance we demonstrated in our manuscript is also beyond that of hybrid-integrated counterparts thanks to the developed 3D integration scheme with low insertion loss, lithographically-defined alignment accuracy and precise on-chip phase tuning. The *linewidth* and *phase noise* are limited by the thermorefractive noise of the resonator, and high *feedback insensitivity* is achieved thanks to the strong Rayleigh back-scattering and ultra-high Q-factor of the ring resonator.

We added a comparison figure (Extended Figure 7) in the Supplementary Information to compare with the results presented in previous demonstrations of heterogeneous integration*. In short, the listed reference articles have not achieved heterogeneous laser integration with ultra-low-loss (< 1 dB/m) SiN waveguides and our results show much better laser linewidth and feedback sensitivity. We thank the reviewer for the suggestion of including a comparison, which will clearly show the superior performance of our demonstrated devices.

*Note: [Guo, Science Advances, 2022] is heterogeneous integrated laser with further hybrid integration and fiber-connected PDH locking. So we didn't include it in the figure but we discussed that the superior performance on laser integrated linewidth could be enabled by our platform to realize the laser integration with ultra-low loss waveguides and micro-fabricated mirror in the future.

We also added a comparison table (Table S2) in the Supplementary Information to compare with the results presented in previous demonstrations of reflection-insensitive lasers. It should be noted here that although we tried our best to make fair comparisons, there has not been a strict standard for the claims on "isolator-free" operation on PICs.

The only standard for fiber-optic systems is IEEE 802.3 standard. A laser source must tolerate at least -21 dB off-chip feedback strength in the fiber-optic system. Nevertheless, this standard is not highly effective for PICs, since the unaccounted fiber-chip coupling loss also determine how much power is sent into the laser cavity, which may vary the concluded feedback insensitivity claims substantially. In this context, we propose a criterion that includes the coupling loss in this paper and focuses on the on-chip feedback strength. Moreover, we used the strictest criterion to determine the reflection insensitivity so far, since **while we increase the feedback strength, we also confirm that the laser linewidth is unaffected by checking the frequency noise performance across a wide offset frequency range.** This criterion helps us to judge the occurrence of feedback regime II in which the laser source is still coherent but linewidth fluctuation might occur. In some related studies on QW lasers [Moehrle, ECOC, 2009; Brac, Nanophotonics, 2021; Matsui, Nature Photonics, 2021], the feedback strength only considers the loss of the off-chip feedback loop, which does not correspond to the case of PIC where on-chip feedback strength is strictly compared. In some related studies on quantum-dot (QD) lasers, one cannot determine if the laser is operating in regime II due to the absence of the frequency noise comparisons [Duan, IEEE PTL, 2019; Dong, PRA, 2021; Dong, Photonics Research, 2021]. Therefore, we believe a comparison of the laser frequency noise with and without optical feedback is critical to determine the stability of laser operation.

We want to highlight that, even if we apply such strict criterion, the reflection insensitivity of our laser is still on par with or even better than the state-of-the-art QD laser which benefits from the low linewidth enhancement factor. The only literature we found that has laser phase noise measurement similar to our work is [Zhang, IEEE JLT, 2020], and their laser exhibited instability under an off-chip feedback strength at -31 dB, while our laser is still unaffected in terms of laser frequency noise with off-chip feedback strength as high as -0.9 dB (Fig. 3e). Such comparison (30 dB difference) clearly shows the superior performance of our demonstrated devices.

Our action taken:

- We added Extended Figure 7 to compare the linewidth results with recent demonstrations of

heterogeneously integrated devices in the Supplementary Information.

- We added Table S2 in the Supplementary Information and related discussions to compare the feedback insensitivity performance with recent representative demonstrations in terms of the gain medium platform, the lasing wavelength, the laser cavity design, and the critical feedback level that destabilizes the laser:

Wavelength	Platform	Design	Critical feedback level		Frequency noise comparison	Ref. No.
			Off-chip	On-chip		
1.55 μm	QW	DFB + high Q resonator	> -0.9 dB	> -6.9 dB	Yes	This work
1.55 μm	QW	DFB	-31 dB	N/A	Yes	[30]
1.49 μm	QW	DFB	-9 dB	N/A	No	[31]
1.55 μm	QW	DFB	-6 dB	N/A	No	[32]
1.56 μm	QW	DFB	N/A	-14 dB [†]	No	[33]
1.59 μm	QDash	FP	N/A	-23 dB [†]	No	[34]
1.31 μm	QW	DBR	-4 dB	N/A	No	[35]
1.3 μm	QD	FP	N/A	> -7.4 dB [†]	No	[27]
1.3 μm	QD	FP	N/A	-3.5 dB*	No	[26]
1.3 μm	QD	DFB	N/A	-6 dB [†]	No	[36]

[†] Only the forward fiber coupling loss is included. Some of these studies defined the boundary of regime IV as the critical feedback level, which excluded the influence of linewidth fluctuation in regime II.

* Back reflection in the short-cavity regime, where the relaxation oscillation frequency is lower than the external cavity frequency.

Table S2. Performance comparison of reflection-insensitive semiconductor lasers.

The heterogeneous integration of this high-Q silicon nitride cavity of 50 million to the active gain material is an impressive demonstration in itself. To achieve this, they have used a low confinement silicon nitride waveguide which allows for less interaction with fabricated surfaces of the waveguide. This platform is excellent for pushing the Q very high, even above 100 million, as shown in their previous works. However, it also requires larger adiabatic mode transitions, larger bends, and a 15 micron oxide underneath. The fabrication process, mode transitions throughout the device, and foundry compatibility of these processes is not discussed at all, so it is difficult to judge their compatibility and this is a major claim of the work. It is also not clear in the manuscript what is the real impact on photonic applications of going beyond 10 million Q on the laser performance. It seems at least the feedback-insensitivity saturates after a Q of about 2 million. The linewidth and noise do

have a strong proportional dependence on the Q, but it would be important to note which applications require such low noise lasers as for most application ~kHz is acceptable.

Our reply:

We thank the reviewer for the comments on the importance of laser integration with ultra-high-Q cavities. We agree that more details on the fabrication process beyond what is already included in the 'Methods' section would be beneficial to a wider audience. We have added a step-by-step process flow charts to present the fabrication details in the Extended Data Fig.3.

- For mode transitions, we also included detailed mode profiles simulations to show the InP/Si, Si/SiN-RDL and SiN-RDL/SiN-ULL hybrid waveguides resulting in highly-efficient 3D mode transitions across the functionality layers. The profile pictures and 3D illustration of the tapers are included in the Supplementary Information Figure 3.
- For the foundry comparability: The foundry compatibility is actually the strength of our approach. The SiN ULL and SiN RDL layer fabrication is performed using a 200-mm diameter process at Tower Semiconductor, a foundry that provides high-performance silicon photonic integrated circuits. We have added related information in the 'Methods' section. While the rest of the process (i.e. the Si and subsequent III-V) are not performed at the foundry yet, we want to highlight that our process is CMOS-compatible and the SiN foundry process shows excellent wafer homogeneity and smoothness for heterogeneous integration. We included additional measurement data to show the heterogeneous integration yield and wafer smoothness in the Supplementary Information (Figure 5). For heterogeneous laser integration, Tower Semiconductor is providing foundry services for InP/Si with OpenLight (PH18DA). They're also doing QD GaAs/Si (PH18DB) with Quintessent now. These progress promises the complete foundry availability of our demonstrated devices in the near future. Moreover, Intel has been commercializing InP/Si process using its own fab and recently Intel has also announced the start of Intel foundry service.
- For the necessity of integrating ultra-high-Q and ultra-low-linewidth lasers: Integrating ultra-high-Q cavities will be critical in expanding the capabilities of silicon photonic integrated circuits. We agree that in many applications, a moderate high-Q around 10 million is already quite capable for many applications and the calculation also shows that the feedback-sensitivity will not be Q-dependent after around 3 million. However, as the reviewer mentions, the laser frequency noise will scale with $1/Q^2$ in self-injection locked lasers and ultra-narrow laser linewidth on the order of Hz or sub-Hz will unlock a lot of applications such as *ultra-low-noise microwave synthesis, precision sensors, atomic clocks*, etc. When used in the context of RF oscillators, the current phase noise (while overkill for optical communications) is still quite high compared to the best compact solutions - quartz crystal oscillators. To beat quartz, the noise still needs to be pushed down significantly to below -160 dBc/Hz at 10 kHz offset. Even higher Q integrated resonators (most likely in addition to optical frequency division) will be necessary to achieve these metrics. The applications of low-noise RF oscillators are numerous, including high-precision GPS-free positioning, navigation, and timing (PNT), next-generation radar, and commercial 5G.

Additionally, an ultra-high Q factor will also benefit on-chip nonlinear processes that require high laser output power for pumping. For example, the Kerr frequency comb generation threshold has quadratic dependence on the cavity Q factor. Cavity Q factors in the ultra-high-Q regime (i.e. > 10 million) will dramatically reduce the laser power requirement for frequency comb generation. This would be extremely critical in enabling highly-efficient multi-wavelength laser frequency comb generators for communications, quantum computing, sensors, etc, which is another example of ultra-high-Q cavity benefiting laser device performances. In addition, mode-locked lasers with narrow wavelength spacing (e.g. Cuyvers, *Lasers & Photonics Reviews*, 2021) could also benefit from < 1 dB/m waveguide loss in the cavity for lower threshold, higher output power and narrower spacing for high-resolution sensing, etc. We added more discussions on the necessity and prospects of integrating ultra-high-Q cavities with lasers in the revised manuscript.

Our action taken:

- We added a process flow chart in the Extended Data Fig.3 to show the fabrication details

- We added mode profile simulations (Supplementary Information Figure 3) to illustrate the mode transitions across the multiple layers

- We added discussion on the foundry capability of our device in the revised manuscript in the Methods (Device fabrication) and Supplementary Information (Surface quality for

heterogeneous bonding). We also added the information that the 200-mm-diameter wafer is “fabricated in a CMOS foundry” in the revised caption of Fig.1a.

- We added discussion on the necessity and prospects of integrating ultra-high-Q cavities with lasers in the revised Supplementary Information: “Integrating ultra-high-Q resonators for ultra-narrow-linewidth lasers with Hertz-linewidth or lower is paramount for applications such as ultra-low-noise microwave synthesis, precision sensors, atomic clocks, and so on. These applications put a more stringent requirement on the laser linewidth than conventional integrated semiconductor lasers could offer.”

The microwave frequency demonstration was quite impressive as the low-noise of the currently demonstrated lasers, despite not being locked to the same cavity, demonstrate phase noise properties much closer to previous integrated frequency comb demonstrations. However, this demonstration is missing some details such as the measurement of the linewidth of the new microwave frequency generation and exact numbers for the phase noise to compare it with state-of-the-art integrated microwave generation. The claim is that now that the lasers are at such low noise levels they can generate microwave frequencies with low noise levels as well, and this does seem true in the demonstration. References to consider explicitly comparing with: Hulme, Optics Express, 2017; Liu, Nature Photonics, 2020.

Our reply:

We thank the reviewer for the positive comments on the microwave signal generation. We agree that a direct comparison with other state-of-the-art integrated photonic microwave generation would help illustrate the difference and performance well. We have included a new comparison table to compare the exact numbers for the phase noise, generated microwave frequency, integration platform, laser integration and laser linewidth in the Supplementary Information (Table S1). We also measured the Gaussian linewidth of the generated microwave signal to be around 15 kHz (Extended Data Fig.2). We want to mention that our heterodyne beat approach has the advantages of widely tunability while maintaining a low phase noise. The demonstrated results show much better signal quality than the only demonstration that achieved the same tunability (Hulme, Optics Express, 2017). Admittedly, the phase noise still falls behind the state-of-the-art integrated frequency comb-based approaches yet (Liu, Nature Photonics, 2020) since there is no phase noise reduction due to the optical frequency division effect in our scheme. However, adopting larger resonators to achieve lower laser linewidth will lead to further improvement in terms of the microwave beat phase noise. Additionally, as pointed out by the reviewer, locking the two lasers to a same cavity will also significantly reduce the phase noise, which we will explore by using the same demonstrated 3D integration platform in a future study. Again, our demonstration has integrated lasers on the same chip, this has not been achieved by any previous low-noise integrated microwave signal generation work. Further improvement on the current device noise performance could be achieved by proper packaging of the chip, which is currently tested by probe cards in a lab environment without shielding from the open air flow.

Our action taken:

- We added a comparison table about photonic microwave generation in the Supplementary Information (Table S1) including the integration platform, approach, frequency, laser integration status, phase noise and laser linewidth:

Platform	Approach	Frequency	Laser	Phase noise (dBc/Hz)	Laser linewidth
SiN [18]	Bright soliton	20 GHz	Off-chip	-106 @ 10 kHz, -130 @ 100 kHz	NA
SiN [19]	Dark pulse	10 GHz	Hybrid	-100 @ 10 kHz, -129 @ 100 kHz	1.2 Hz
III-V/SiN [14]	Mode-locked laser	755 MHz	On-chip	-85 @ 10 kHz, -108 @ 100 kHz	200 kHz
III-V on Si [20]	Mode-locked laser	20 GHz	On-chip	-67 @ 10 kHz, -85 @ 100 kHz	NA
III-V/Si [21]	Heterodyne	0-50 GHz (Tunable, PD limited)	On-chip	NA	150 kHz
This work	Heterodyne	0-50 GHz (Tunable, PD limited)	On-chip	-53 @ 10 kHz, -83 @ 100 kHz	5 Hz

Table S1. Comparison of reported photonic microwave generation schemes.

The demonstration in the manuscript is impressive, but is missing important details which would place the work in proper context and show whether this is truly a major achievement over the most recent demonstrations.

Our reply:

We agree with the reviewer that more explicit comparison would be helpful for the readers to understand our achievements better. We have included useful contents in the revised manuscript, Supplementary Information and we believe the following messages are clearer: (i). Despite overcoming significant integration challenges, the demonstrated devices show higher integration level and better laser linewidth results than hybrid-integrated counterparts. (ii). The feedback insensitivity also shows about at least 30 dB improvement over previous studies using the same standards. (iii). The widely tunable microwave generation using integrated lasers is demonstrated for the first time with low phase noise thanks to the attained ultra-low laser frequency noise.

For that reason, I recommend the authors must address the concerns above and the following additional points in a future revision to be accepted:

1. There should be a measurement of the quality factor of the ring resonator as one of the major claims is that the noise and feedback-insensitivity is strongly dependent on this.

Our action taken:

We added the quality factor measurement results in the Extended Figure 1:

2. How are the losses measured in Fig. 1?

Our reply:

The loss is extracted from the transmission sweep of the ring resonator resonances and after obtaining the fitted intrinsic quality factors using the formula: $\alpha = \frac{2\pi \cdot n_g}{Q_0 \lambda}$, where α is the loss within the ring resonator, n_g is the group index, Q_0 is the intrinsic quality factor and λ is the wavelength.

Our action taken:

- We added explanations on the method of loss and Q measurement in Extended Data Fig.1a caption and revised Fig.1 caption “loss extracted from the fitted resonator Q factors”.

3. How was the linewidth of the laser measured? The measurement and details are missing. This should have measurement error statistics.

Our reply:

The laser linewidth is measured using the OEwaves OE4000 phase noise analyzer. As an automated equipment it is fully capable of rapidly measuring laser phase noise without complex setup. It is based on a homodyne methodology, so no reference lasers are required to make such a narrow linewidth measurement. The full details are not available to us yet since this is a commercial equipment. However, we also conducted linewidth measurement using custom-built setup based on cross correlation techniques when we tested other lasers. The measurement shows excellent agreement with the results obtained using OEwaves OE4000. We added a section in the Supplementary Information to discuss the linewidth measurement methods.

For linewidth measurement errors, we want to point out that, we have optimized the phase conditions for the phase noise measurement shown in the Fig.2e. As it is now completely limited by the microresonator thermorefractive noise (TRN) floor, the measurement results are very stable and the standard deviation was negligible for several measurement runs. In addition, when taking the phase noise trace, we have already used the Internal “Average” option provided by the OE4000 equipment to do averages (5 to 50 times) which helps to reduce measurement errors to negligible amounts.

Our action taken:

- We added explanations on the linewidth measurement in the ‘Methods’ section with details on the averaging and alternative methods
- We discussed more techniques for the laser frequency noise and linewidth measurements in the Supplementary Information (section “Methods to measure the laser frequency noise”).

4. What would be the theoretical limit of the noise and linewidth based on Q’s that can be achieved on this platform.

Our reply:

The phase noise and linewidth will scale as the noise reduction factor (NRF) which is dependent on the cavity Q factor at the strong back-scattering regime: $NRF \approx 4(1 + \alpha^2)T^2\eta^2\frac{Q_R^2}{Q_d^2}$, where α is the linewidth enhancement factor, T is the laser to resonator coupling ratio, η is the bus waveguide to ring resonator coupling coefficient, Q_R and Q_d are the quality factor of resonator (loaded) and laser diode respectively. Using $\alpha = 2.5$, $T = 80\%$, $\eta = 0.5$, $Q_d = 10^4$, $Q_R = 50 \times 10^6$, we can calculate that the NRF is as high as 80 dB. As a result, the current phase noise is completely limited by the TRN and the current NRF is about 50 dB (from ~ 800 kHz free running DFB fundamental linewidth to ~ 5 Hz after locking). A conservative estimation is that extending the ring radius to 100 times larger to account for extra 20 dB noise reduction would result in laser linewidth on the order of 10 mHz which has been demonstrated in a hybrid-integrated format (40 mHz) [Li, Optics Letters, 46, 5201, 2022]. It needs to be noted the low laser to resonator coupling loss plays a critical role to enable lower linewidth lasers for our 3D integration scheme compared with hybrid-integration. To illustrate the potential of this platform, we included another plot in the Supplementary Information to calculate the achievable laser linewidth.

Our action taken:

- We added a discussion section in the Supplementary Information (“Discussion on laser linewidth improvement”) to show the achievable laser linewidth on our platform and the relative limitations from the resonator diameter and advantages in low laser to resonator coupling loss.
- We added a plot in the Supplementary Information (Supplementary Figure 8) to show the achievable laser linewidth as a function of the waveguide loss and its limitation with TRN floor.

5. It is difficult to understand the full mode transition of the 3D vertical transition from the active gain material of the laser to the high-Q silicon nitride layer. All the mode transitions and fabrication processes should be included in the supplementary.

Our reply:

As we replied previously, we have updated the Methods section, Extended Data Figure and Supplementary Information to include more details.

Our action taken:

- We included more description of the fabrication process in the methods section ‘Device fabrication’
- We added the mode transitions and the fabrication process flow charts in the Supplementary Information Figure 3 and Extended Data Fig.3 respectively.

6. The high-Q shown here can also be achieved in a high confinement sin platform that does not create such large mode sizes, are there advantages or disadvantages that pertain to the noise and feedback-sensitivity characteristics of the laser in high confinement vs. low confinement cavities?

Our reply:

The high confinement SiN platform is important in enabling dispersion engineering for nonlinear process including dissipative Kerr bright soliton. The demonstrated Q factor, however, is normally lower than the low-confinement SiN platform that we adopted here. As the feedback sensitivity and laser noise has strong dependence on the Q factor, the low-confinement platform has advantages in enabling high feedback insensitivity and low laser phase noise. Indeed, the demonstrated 3D integration techniques can also be applied to high-confinement SiN platform to maintain the high-Q factor during laser integration process for the best performance.

Our action taken:

- We added the discussion in the revised manuscript on the potential of extending our integration techniques to high-confinement SiN platform to achieve the optimized performance: “Our platform could also be used with thick SiN waveguide layers with tight mode confinement for nonlinear applications that require both anomalous dispersion and high Q-factor cavities [25,26]”.

7. A measurement of the linewidth of the microwave frequency generated is missing and a calculation or description of how this is quantitatively related to the laser properties.

Our reply:

We thank the reviewer for this comment. Indeed, we think the phase noise of microwave signal reveals more properties more than a linewidth measurement as it shows the signal noise dependence on offset frequency. For different applications, the critical offset frequency varies and the phase noise characterization shall contain more information than a linewidth value for the microwave frequency. As the linewidth can be extracted from the phase noise, the microwave linewidth/phase noise are both directly related to the laser linewidth/phase noise. In theory the microwave signal phase noise is the sum of the noise from the two heterodyne beating lasers. If lasers with similar phase noise are used for the heterodyne beating, the phase noise of microwave signal would be around 3dB higher than individual lasers. Nevertheless, we added an RF spectrum of the heterodyne beat between the two lasers in the Extended Data Fig.2.

Our action taken:

- We added an electrical spectrum of the microwave frequency signal with Voigt fit and extracted the Gaussian linewidth to be 15 kHz (Extended Data Fig.2):

- We discussed the relationship of microwave phase noise and linewidth dependence on the laser properties in the revised manuscript: “In such heterodyne beating schemes, the generated microwave signal phase noise is the sum of the phase noise of the heterodyne beating lasers.”

8. How is the R value, feedback fraction, calculated? Can this be compared with optical isolator specifications that would be needed to achieve the same performance?

Our reply:

There are two values related to the reflection values. The first is the ‘R value’ in Fig.3c and another is the applied on-chip feedback strength (feedback fraction) which is calculated using the summary of fiber-chip coupling loss, external fiber-based variable optical attenuation and fiber system loss. Details are as follows:

- The R values shown in Fig. 3c is the back scattered reflection from the high-Q ring resonator into the bus waveguide. This R value is what is measured and extracted in the added Extended Data Fig.1b. We included a section in the Supplementary Information discussing the method for the reflection value extraction from a resonance transmission sweep and fitting.
- Fig. 3d discussed the applied equivalent on-chip feedback strength that is controlled using an off-chip back-reflector and variable optical attenuator. The feedback strength is determined by the reflected power P_{refl} and the free-space output power P_{out} through the relationship $\eta_F = \frac{P_{refl}}{P_{out}}$. All losses from the feedback loop are considered (including the coupling loss and the off-set component insertion loss, etc) to correctly calculate the reflected power. The details of the calculation can be found in the “Methods” section, under “Laser feedback sensitivity

measurement”. We also modified the top axis label in Fig. 3d to be ‘Applied on-chip feedback strength’ to avoid misunderstanding.

To compare with optical isolator specifications that would be needed to achieve the same performance, a fair method is to consider the difference between the free-running and self-injection locked states as shown in Fig. 3d. As the SIL allows for a >34 dB and 27 dB improvement of the feedback insensitivity at the drop port and through port, which means that an effective isolation of more than 34 dB and 27 dB have been achieved to maintain the original laser coherence.

Our action taken:

- We modified the top axis label of Fig. 3d to be “Applied on-chip feedback strength (dB)” to make the message easier to understand
- We added discussion on the “effective isolation” in the revised main manuscript: “Such 27 dB and over 34 dB improvement in the feedback insensitivity are equivalent to the effective isolation that optical isolators can provide to maintain the laser coherence”.

9. How does the feedback insensitivity of the lasers affect the microwave frequency generation?

Our reply:

The feedback insensitivity of lasers results in the isolator-free operation of the microwave frequency generation scheme. We compared the microwave frequency measurement with and without isolators. The generated microwave signal shows same phase noise. We have already mentioned in the main text about the isolator-free operation of microwave generation experiment: *‘The advantage of feedback insensitivity is also critical in direct on-chip microwave synthesis since several components including the 3-dB couplers and photodiodes need to follow the lasers and are potentially strong sources of on-chip reflection’* and the figure 4 caption: *‘b. Experimental setup for the isolator-free, widely-tunable heterodyne microwave generation’*.

10. It is mentioned that increasing the resonator cavity radius can reduce the thermorefractive noise. Would this imply a fundamental tradeoff between the mode-hop free tunability of the laser and noise?

Our reply:

The resonator thermorefractive noise will scale with cavity radius/circumference. A larger cavity radius would result in lower thermorefractive noise, but narrower free spectral range (FSR). Indeed, ring resonator based external-cavity Vernier lasers with narrow ring FSR would be very easy to mode hop due to the narrow longitudinal mode spacing. However, in the current scheme, we rely on self-injection locking of DFB lasers with single longitudinal mode to match the ring resonances for locking. That said, the laser wavelength tuning is achieved by the tuning of DFB laser output wavelength (through laser gain current tuning) and ring resonance tuning (through the heater current tuning). In order to achieve mode-hop-free tuning, the laser wavelength and the ring resonances need to be tuned at the same rate, which is independent of the resonator diameters or FSRs. So, extending the ring resonator radius would not make a difference in the mode hop tuning property of our self-injection locking based schemes.

Our action taken:

- We added discussion about the mode hop tuning in the Supplementary Information: “Increasing the resonator radius/circumference to reduce the TRN will lead to lower FSR. It needs to be noted that, in ring-resonator-based Vernier lasers, extended ring circumference can lead to narrow longitudinal mode spacing that gives rise to severe mode hops. For the self-injection locking scheme in our work, the mode-hop-free tuning of the lasing wavelength can be achieved by the simultaneous tuning of the DFB lasing wavelength by the gain current tuning and the

ring resonance by heater tuning at the same rate. Such requirement is independent of the resonator diameter or FSR. As a result, the increase of the ring circumference will not increase the chance of mode-hops as in the case of Vernier ring lasers”.

11. Fig. 2e should include noise floor of the instrument as well.

Our action taken:

- We added the noise floor of phase noise analyzer we used (OEwaves OE4000) to Fig. 2e to show the margins.

12. Last main text figure has a letter labeling typo.

Our reply:

We thank the reviewer for spotting the typo. We have updated the labelling description in the Fig.4 caption.

Our action taken:

- We corrected the description of Fig. 4 in the figure caption.

Referee #2 (Remarks to the Author):

The authors demonstrate an isolator-free self injection locked laser platform based on 3D integration. They demonstrate 3D integration of lasers with low loss waveguides, self injection locking of the lasers, isolator free operation, and tunable microwave frequency generation. The work is very well written and clearly presented. The manuscript is technically sound and aside from a minor issue with figure 4 is excellent. My main concern is that, in my opinion, the manuscript seems too specialized, as presented, for Nature. The manuscript goes in depth into the injection locking and the noise measurements. Additionally, the authors have presented significant results in prior publications on this topic. The main advance seems to be bringing these prior innovations together into one device.

Our reply:

We thank the reviewer for the positive evaluation of both quality and depth of our demonstration and presentation. We want to emphasize that, our manuscript includes contents on the novel device design and fabrication, laser locking dynamics investigation, device performance characterization (laser phase noise, feedback insensitivity and microwave generation) and related theoretical analysis. We believe it is a quite broad in-depth study that can be helpful for research communities with diverse background in photonic devices, laser physics, and applications that range from microwave synthesizers and communications to quantum sensors and atomic clocks, etc. Besides, the 3D integration concept, which is showing its significance in semiconductor industry for advanced IC functionality and density, is being used together with ultra-low-loss silicon photonics and III-V photonics for the first time in our work. Moreover, the demonstrated integration architecture holds great promise to be further 3D integrated with electronic ICs for 3D E-PICs and solve important issues on both ends of electronics and photonics including interconnect bandwidth, driver capacitance and so on. As such, we believe our results can play an important role in various fields rather than being a specialized study.

The reviewer also expressed opinions about the significance of our work when comparing to prior ‘hybrid-integrated’ demonstrations that uses multiple chips. While we thank the reviewer to acknowledge our achievements in “bringing innovation on a single chip” – which in our opinion is a challenging and already significant task, we want to explain in detail that, our demonstration represents significance over the prior publications of hybrid-integration in several aspects:

1. Moving from multiple chips to one device is not only a single attempt to avoid optical packaging. As the reviewer mentioned, our work brought ‘these prior innovations together into one device’. This is indeed true and is indeed a major challenge that used to prevent the silicon photonic integrated circuits (PICs) from further improving the device complexity and scalability due to missing on-chip components. For example, heterogeneous integration of III-V materials on a single Si PIC offers the laser gain and amplifier gain that allow dense laser integration and direct on-chip amplification for large scale PICs. The meaning is beyond just simply moving several chips into a single chip, since the PIC architecture can be dramatically different between a multi-chip package that relies on optical coupling and a single-chip that supports complex waveguide routing, on-chip source, amplifiers, and on-wafer testing.
2. Single chip has additional advantages over multi-chip packages in many ways:
 - a. Single-chip integration allows wafer-scale production and integration that would result in lower cost and higher scalability. It also avoids the time-consuming and expensive optical packaging which would normally take up to 70-80% of the total cost.
 - b. Single-chip integration permits robust device interconnects and offers the highest stability over time. The optical phase can be precisely controlled without being influenced by the free space gap between multiple chips causing power or phase fluctuations. This type of vulnerability is quite sensitive in ultra-low-noise applications.
 - c. Single-chip integration demonstrated in our work leverages advanced photolithography for the fine alignment across different layers (< 100 nm misalignment) and the resultant low inter-layer transition loss. This would avoid the high coupling loss that normally exists in hybrid packaging associated with degraded self-injection locking strength and effects. Our result shows lower phase noise than hybrid-integrated self-injection locked lasers with the same 30 GHz resonator, which is a good indication of such advantages.
3. Enabling single chip integration of optical functionalities faces significant difficulties. As we have presented in the manuscript, during heterogeneous single-chip integration, the process compatibility and underlying realities for the optimized performance of different functionality materials needs to be addressed carefully. We used 3D integration scheme to solve the problem of loss performance degradation during the process and improved the device yield using optimized semiconductor fabrication techniques. So, our result represents critical advances in terms of both designs and fabrication of complex PICs which would not be possible with multiple-chip based hybrid integration.

Moreover, we want to highlight that, while enabling the challenging single-chip integration is a major accomplishment in our manuscript, we also demonstrated several tasks that have not been successful by hybrid-integrated devices so far and show the intrinsic advantages of our schemes:

1. The feedback sensitivity is a major issue for the laser integration with downstream optical components that might result in destabilized laser working status. For hybrid integration that relies on multi-chip packaging, there are possibilities to use inserted optical isolators for the purpose of optical isolation between the laser and downstream components which would however introduce higher cost and additional complexity. For single-chip integration, due to the difficulty of integrating magnetic materials in a CMOS fab for on-chip isolators, it is difficult to isolate the laser from downstream on-chip components, so the laser feedback insensitivity is of critical importance. We provided practical solutions using the 3D integration of laser with ultra-high-Q resonators to increase the feedback insensitivity by at least 30 dB more than previous studies using high-Q cavities to reduce feedback sensitivity. Besides, there have not been any reported results of hybrid integration for feedback insensitivity yet.
2. We reported the microwave generation using two of the self-injection locked ultra-low-noise laser on the same chip for widely-tunable microwave signal generation. This has only been reported once [Hulme et al, Optics Express 2017], but this previous study used an integrated laser with linewidth of 150 kHz which is around 5 orders of magnitude noisier than what we have demonstrated in this work. We believe our work represents a major progress in this heterodyne

photonic microwave synthesis scheme. This scheme, thanks to the dense integration of heterogeneous integration allows the lasers and resonators to be integrated on the same chip, offers significant advantages over hybrid integration in terms of the device footprint, overall stability, and potential to further integrate photodetectors for fully-integrated direct on-chip microwave signal synthesis. In addition, no hybrid integrated devices have been demonstrated for this low-noise widely-tunable heterodyne microwave signal generation yet. As a result, our results show both advantages and novelty in this important application.

3. We carried out detailed investigation on the phase dependence of self-injection locking dynamics and identified, for the first time, the existence of locking, free-running and chaotic states depending on the relative phase relations between the forward/backward in such laser-resonator pairs. To precisely control the optical phase using on-chip phase tuners offers the capabilities for such studies. Similar research is difficult to implement with hybrid-integrated devices that rely on free space coupling for phase tuning, which is not only inaccurate but also causes power and feedback strength fluctuations that would affect the discoveries on pure phase dependence.

In sum, Si photonics needs such fully integrated single-chip lasers and our results show the unique advantages over multi-chip packages. We thank the reviewer for the comments, and we want to draw the reviewer's attention to our abstract which clearly discussed the motivation for the pursue of such single-chip devices. In our main manuscript, there are discussions on the advantages of enabling single-chip integration:

- *“Due to the availability of an on-chip phase tuner between the laser and ring resonator, we can clearly unveil the phase-dependent locking dynamics. In previous butt-coupled SIL experiments, tuning the chip-to-chip phase also varies the coupling loss, i.e. the output power. Because the InP/Si laser and SiN resonator are heterogeneously integrated together and the phase is thermally tuned on chip, these are now decoupled in our experiment.”*
- *“The capability of integrating ultra-low-noise lasers at the wafer scale opens up the possibility of enabling photonic devices that were impractical to integrate.”*
- *“no isolators are used in the experiment, which shows the feedback insensitivity could greatly simplify the system architecture and permit a fully on-chip integrated microwave synthesizer when couplers and PDs are integrated on the same 3D Si PIC”*

We have also included more discussions and comparison in the revised manuscript to explain the significance of our work more clearly.

Our action taken:

- We added in the revised main text “Here, we combine monolithic and heterogeneous 3D integration to fully unlock the potential in enabling complex and high-performance photonic devices and integrated circuits” to emphasize the difference of our work from the multi-chip packaged hybrid-integrated devices
- We added more details in the device fabrication, design and characterization to illustrate the challenges we have overcome to demonstrate the single-chip device. These details can be found in the Extended Data figures (1-3) and Supplementary Information (Supplementary Figures 2,3,5,7,9,11).
- We added the comparison of our device with other reported devices in terms of linewidth performance, feedback sensitivity and microwave generation performance. We clearly highlighted the strength of 3D single-chip integration in these comparisons (Supplementary Figure 7 and Table S1, S2)
- We added the discussion on that using single-chip integration for lower coupling loss is critical in enabling stronger self-injection locking effect in the Supplementary Information: “The low laser-to-resonator coupling loss benefits the linewidth reduction compared with hybrid integration schemes that normally suffer from a high coupling loss”

Minor Concern:

- In figure 4, there appears to be a mismatch between the text, the figure, and the figure caption. The figure only includes parts a through d, while the caption mentions part e. Based on the figure caption it seems to me that the authors meant to label the beat frequency vs time plot as part c. This should be corrected.

Our reply:

We thank the reviewer for spotting the typo. We have updated the label description in the Fig.4 caption.

Our action taken:

- We corrected the description of Fig. 4 in the figure caption.

Referee #3 (Remarks to the Author):

The work reports a multi-layer approach for the heterogeneous integration of ultra-low-linewidth lasers in silicon photonics. Two layers of ultra-low-loss silicon nitride are stacked on top of each other, followed by a silicon photonic PIC and III-V gain chiplet. The lasers are implemented by injection locking a distributed feedback laser to a high-Q optical microresonator cavity. The DFB laser is implemented in the Si-III-V stack, taking advantage of the active gain provided by the InP material, and the ultra-high-Q cavity is implemented in the bottom SiN layer.

Although similar 3D integration approaches have been developed in recent years, as the authors recognize in the reference list, this is the first time to my knowledge that the SiN layer displays losses in the dB/m range. This accomplishment is crucial for attaining a sufficiently high-quality factor in the resonator used for the external cavity and reach an integrated laser with excellent coherence performance. Critical to this is the development of the intermediate SiN layer, which helps to bury the ULL layer at the bottom, far from the semiconductor stack, and minimize impurities that would otherwise degrade the loss performance. The redistribution layer also assists in the realization of presumably efficient transitions between the distinct layers and materials.

Another important aspect of the study is the resilience of the laser architecture to spurious reflections. The authors do a remarkable analysis of the laser dynamics and show that with sufficiently high Q, the laser could withstand large reflections without affecting the injection locking to the cavity nor the purity (coherence). The significance of the architecture, as the authors recognize, is that the laser avoids the use of optical isolators, which are complicated to manufacture and typically require materials that are currently not compatible with the CMOS processing lines.

The content of the manuscript is extremely well organized. The figures are clear and succinct, and I find an excellent balance between technical detail, forward-looking discussions and rigor.

In my opinion, the achievement of heterogeneous integration of ultra-low-linewidth lasers in silicon photonics deserves the visibility that a publication in Nature provides.

Our reply:

We thank the reviewer for the positive evaluation of our work.

I only have a couple of remarks that I warmly encourage the authors to consider for clarifications:

1. The introduction gives at instances the false impression that this is the first time that a 3D integration approach in silicon photonics is reported including active devices. However, references 32, 34 and 35, just to take some from the reference list, have developed similar multilayer integration

techniques that include active components. I believe the manuscript achieves for the first time a multilayer integration approach between actives and passives that preserves the ultralow losses in the SiN layer, but this message is not clearly conveyed in the introduction, hence decreasing the value/merit of the previous studies in the field.

Our reply:

We thank the reviewer to help point it out. We agree that we should have been more accurate in the description of the 3D integration reported in our work. The original reference 32 [Sacher et al, Proceedings of IEEE, 106(2232), 2018] demonstrates *monolithic 3D* Si devices including modulators and photodetectors integrated with multiple deposited SiN layers on SOI waveguides. Original references 34 and 35 [Xiang et al, Optica, 7(20),2020 and Xiang et al, Science 373(99), 2021]] are multilayer *heterogeneous* integration that enabled laser integration with SiN waveguides and SiN frequency comb generators. Here in our work, we combined the *monolithic* and *heterogeneous* 3D integration to facilitate the III-V laser integration with Si and ULL SiN photonic circuits. Compared with these demonstrations, a unique difference in our current manuscript is the use of SiN redistribution layer to control the interlayer coupling that could not be achieved with high efficiency between two layers (Si and SiN ULL) with large vertical spacing (See Supplementary section I “3D layer transition design”). The large vertical spacing is critical in maintain the ULL of SiN during the fabrication process. The reviewer is correct that such design and architecture enabled the first-time ultra-low-loss (< 1 dB/m) waveguide integration with active III-V materials for lasers. We updated the manuscript in related discussions and highlighted the difference of our demonstration with previous studies.

Our action taken:

- We updated the discussion about 3D integration in the revised manuscript to make it clearer: “In photonics, 3D integration has been investigated for monolithic devices (e.g. waveguides, modulators, and photodetectors) [30], and heterogeneously-integrated lasers [31]. Here, we combine monolithic and heterogeneous 3D integration to fully unlock the potential in enabling complex and high-performance photonic devices and integrated circuits

2. The injection locking technique is now well established and relies on the Rayleigh scattering in the high-Q microresonator coupling forward and backward modes. This process is non-controllable, raising concerns over how repeatable the design is and what is ultimately the fabrication yield. The analysis in Figure 3c indicates that in the high Q regime, there is a large range of scattering coefficients that would result in a laser design that is resilient to reflections $< 10\%$, but still, how likely is it to get a scattering coefficient with $R > 0.25\%$? Does this happen for all wavelengths, i.e., do all resonant modes display a split resonance profile? What are the implications on the laser yield?

Our reply:

We agree with the reviewer that the Rayleigh scattering in high-Q microresonator is a non-deterministic process, originating from sidewall roughness incidental to the fabrication process. However, while each individual resonance exhibits a variable amount of back-reflection, the ensemble average of all resonances within the resonator will exhibit a repeatable average splitting within a given fabrication process. Among 505 resonance spectra measured within the S+C+L bands for this device, we infer that 475 (approximately 95% of resonances) exhibit a peak reflectivity $R > 0.25\%$. The histogram data in added Extended Data Fig.1c shows that the most probable peak reflection R is around 3%. Thus, every laser device is assured to be capable of self-injection-lock throughout the S+C+L bands, with almost complete coverage of resonance wavelengths. We note that additional wavelength flexibility can be provided by using the integrated heater to fine-tune a resonance to a desired wavelength.

We note that the absence of obvious splitting in the transmission spectrum line shape does not imply a low back-reflection. In the Extended Data Fig.1a we show resonances with and without splitting,

measured in the same resonator. Both resonances show substantial reflection that is sufficient for self-injection locking.

Additionally, future work could use grating structures (e.g. [Yu et al. Nat. Photonics 15, 461–467 (2021)]) within the high-Q ring resonator to further increase and control the back scattering for the laser locking.

Our action taken:

- We added a figure in the Extended Data to show the extracted R values from the fitting the resonance lineshape, for the same data set as presented in Figure 1b in the main text. The red area marked in the Figure c panel shows the resonant mode with scattering coefficient R below 0.25%, which represents 30 of the 505 resonances analyzed.

- We added discussion on the implications of high reflection of resonant modes on the self-injection locked laser yield in the Extended Data Fig. 1: “While the reflection coefficient is variable between resonances, a majority of resonances exhibit sufficient reflection for self-injection locking. Considering the capability of resonance tuning, the resonance reflection would not affect the self-injection-locked laser yield” and “More control on the back reflection strength could be implemented by leveraging grating structures within the ring resonator [59].”

3. The fabrication details are quite slim. For example, is CMP only implemented after the deposition of the second layer? What is the homogeneity and roughness of the films? What strategy is used to properly align the masks? Is annealing implemented at some step to achieve ultralow losses? If so, under what conditions?

Our reply:

We agree with the reviewer that more details about the fabrication process flow can be very helpful. Here are our answers to the reviewer’s questions about fabrication:

1. CMP. There are two CMP steps. First CMP step is implemented after the first layer SiN deposition plus SiO₂ cladding deposition to prepare a flat wafer surface for the second SiN layer deposition. After the second SiN waveguide layer formation and cladding deposition, the second CMP step is performed to prepare a smooth wafer top surface for the subsequent SOI

bonding. The CMP creates a highly-uniform and smooth wafer top surface that enabled a high bonding yield as shown in the Supplementary Fig. 5a.

2. Film quality. We supplemented two plots including the long-range wafer surface topography measurement by a surface stylus profilometer (Bruker DektakXT) and top surface roughness after the CMP measured by atomic force microscopy (Bruker ICON). The measurements are performed on the SiN wafer after the CMP process before subsequent SOI bonding process. The data shows excellent flatness and smoothness of the films (Supplementary Fig. 5b, c).
3. Mask alignment. The two SiN layers are fabricated in a CMOS foundry using a Cannon 248-nm DUV Stepper on a 200-mm-diameter wafer. The 200-mm-diameter wafer was subsequently cored into two 100-mm-diameter wafers for processing at the UCSB Nanofabrication Facility using a ASML 248-nm DUV stepper. All lithographic alignments are performed using alignment marks patterned in the first SiN layer at the beginning of the process. Alignment marks in DUV stepper tools generally leverage phase-grating-based wafer alignment technology and are provided by the manufacturer. These tools routinely deliver < 100 nm registration accuracy between layers.
4. Annealing. High temperature annealing is performed after the SiN waveguide layer etch and SiO₂ cladding deposition identical to our prior work on ultra-low-loss SiN waveguides [Jin et al Nature Photonics, 15, 346, 2021]. The purpose is to drive off the hydrogen content during the SiN and SiO₂ deposition. The total annealing time exceeds 20 hours at 1150° C.

Our action taken:

- We added the above information and other fabrication details in the updated Methods section of our manuscript, including a reference to refer readers for more details of the low-loss SiN foundry process.
- We added step-by-step process flow chart to convey more information about the fabrication process in the Extended Data Fig. 3.:

- We added the topography measurement and surface roughness measurement and a picture showing the high bonding yield (> 95%) in the Supplementary Information Fig. 5.

4. The analysis of the 3D transitions would benefit from clearer illustrations. Also, are the transition losses stated in the manuscript measured or calculated? I appreciate it is challenging to conduct measurements of the crosstalk and losses, especially when the transitions are very efficient, but it would be helpful to share the data with the community if available.

Our reply:

We agree with the reviewer that the detailed mode analysis is beneficial to illustrate the mode transitions better. The overall transition loss includes several stages while the InP/Si and Si/SiN taper losses values were taken from previous studies with same coupler designs [Davenport, IEEE JSTQE 22(78), 2016; Bauters, Optics Express 21(544), 2013; Xiang Nature Communications 12(6650), 2021]. In addition to these prior works, the new taper in this work is the SiN RDL to SiN ULL taper, which has about 4 um thick separation between the layers. We were unable to characterize the precise transition loss on the fabricated laser device as it is very low. However, a dedicated test device, consisting of two SiN-to-SiN layer transitions formed into a single resonator, was fabricated on a separate wafer with somewhat narrower spacer layer (approximately 3.5 um), and measured to exhibit approximately 0.03 dB of insertion loss per layer transition. We thus conservatively expect the insertion loss of the RDL-to-ULL transition within the heterogeneous laser to be well below 1 dB.

Our action taken:

- We have included measurement data (in Supplementary Figure 2) and a discussion of the SiN-to-SiN layer transition device to the supplementary information:

- We have included a figure illustrating the evolution of the mode as a 2D cross-section at various locations through the taper transition structure (Supplementary Figure 3)

Reviewer Reports on the First Revision:

Referee #1 (Remarks to the Author):

The authors have addressed all of my questions and concerns with extensive quantitative comparisons with previous works, new experimental data, and added discussion and text within the main text and supplementary information. The ability to integrate more than one laser heterogeneously with a low loss photonic platform is a challenging feat in itself and an important step forward for integrated photonics. In addition, the laser performance is state-of-the-art in terms of linewidth and noise characteristics making the demonstration quite exciting. I recommend the manuscript is accepted without further revisions.

Referee #3 (Remarks to the Author):

The authors have perfectly addressed all my questions and concerns. I am happy to recommend the publication of the manuscript in Nature.